# Rectified Gradient: Layer-wise Thresholding for Sharp and Coherent Attribution Maps

## Abstract

Saliency map, or the gradient of the score function with respect to the input, is the most basic means of interpreting deep neural network decisions. However, saliency maps are often visually noisy. Although several hypotheses were proposed to account for this phenomenon, there is no work that provides a rigorous analysis of noisy saliency maps. This may be a problem as numerous advanced attribution methods were proposed under the assumption that the existing hypotheses are true. In this paper, we identify the cause of noisy saliency maps. Then, we propose **Rectified Gradient**, a simple method that significantly improves saliency maps by alleviating that cause. Experiments showed effectiveness of our method and its superiority to other attribution methods. Codes and examples for the experiments will be released in public.

## 1 Introduction

The gradient of the score function with respect to the input, also called the saliency map (Erhan et al., 2009; Baehrens et al., 2010; Simonyan et al., 2014), is the most basic means of interpreting deep neural networks (DNNs). It is also a baseline method for other advanced attribution-based methods. However, our understanding of saliency maps is still poor.

Previous studies such as Springenberg et al. (2015) and Selvaraju et al. (2017) have noted that saliency maps tend to be visually noisy. To explain this phenomenon, Sundararajan et al. (2016) and Smilkov et al. (2017) suggested saturation and discontinuous gradients as the causes (see Section 2.1 for further explanation). There were several studies attempting to improve saliency maps by tackling these hypothesized causes (Bach et al., 2015; Montavon et al., 2017; Sundararajan et al., 2016; Shrikumar et al., 2017; Smilkov et al., 2017; Sundararajan et al., 2017).

Even though such attribution methods generally produce better visualizations, we find troubling that the hypotheses regarding noisy saliency maps have not been rigorously verified (see Section 2.2 for more detail on attribution methods). In other words, numerous attribution methods were built upon unproven claims that gradient discontinuity or saturation truly causes saliency maps to be noisy. This situation gives rise to two major problems. First, if the hypotheses regarding noisy saliency maps are incorrect, current and future works based on those hypotheses will also be erroneous. Second, as we do not know precisely why saliency maps are noisy, we have to rely on heuristics and guessworks to develop better attribution methods.

In this paper, we address these problems by identifying saliency maps are noisy because DNNs do not filter out irrelevant features during forward propagation. We then introduce **Rectified Gradient**, or RectGrad in short, a simple technique that significantly improves the quality of saliency maps by alleviating the cause through layer-wise thresholding during backpropagation. Finally, we demonstrate that RectGrad produces attributions qualitatively superior and quantitatively comparable to other attribution methods. Specifically, we have the following key contributions:

- We explain why saliency maps are noisy. Noise occurs in saliency maps when irrelevant features have positive pre-activation values and consequently pass through ReLU activation functions. This causes gradients to be nonzero at unimportant regions. We perform experiments with networks trained on CIFAR-10 to justify our claims (Section 3).

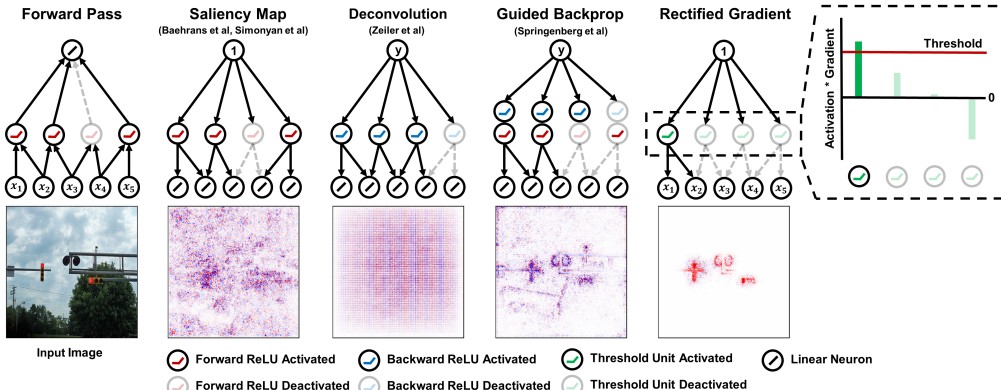

Figure 1: Comparison of attribution methods. See Section 5 for details on the visualization.

- We introduce Rectified Gradient, a method that removes noise from saliency maps by thresholding irrelevant units at ReLU binary gates during backpropagation (Section 4). We first explain the rationale behind Rectified Gradient (Section 4.1). We then prove that Rectified Gradient generalizes Deconvolution and Guided Backpropagation (Section 4.2). In addition, we discuss two techniques that enhance the visual quality of Rectified Gradient attribution maps (Appendix C).

- We first investigate the effect of threshold level on attribution maps produced by Rectified Gradient (Section 5.1). Then, we apply Rectified Gradient to networks trained on CIFAR-10 and ImageNet to demonstrate that it produces qualitatively superior attribution maps (Section 5.2). We also compare Rectified Gradient with other attribution methods using several quantitative metrics (Section 5.3).

## 2 BACKGROUND OVERVIEW

Let $S : \mathbb{R}^d \mapsto \mathbb{R}^{|C|}$ be an image classification network, where $x \in \mathbb{R}^d$ is a single image instance and $C$ is the set of image classes. Then, we can define a score function $S_c : \mathbb{R}^d \mapsto \mathbb{R}$ for each class $c \in C$ and the final class of the image $x$ is given by $class(x) = \arg\max_{c \in C} S_c(x)$. A typical score function is constructed by alternately composing affine transformations and nonlinear activation functions. A squashing function such as softmax is applied to the final layer.

Since functions comprising $S_c$ are differentiable or piecewise linear, the score function is also piece-wise differentiable. Using this fact, Erhan et al. (2009), Baehrens et al. (2010) and Simonyan et al. (2014) proposed the saliency map, or the gradient of $S_c$ with respect to $x$, to highlight features within $x$ that the network associates with the given class. In an ideal case, saliency maps highlight objects of interest. However, previous studies such as Springenberg et al. (2015) and Selvaraju et al. (2017) have pointed out that saliency maps tend to be visually noisy, as verified by Figure 1. Three hypotheses were proposed to account for this phenomenon. We describe them in the next section.

### 2.1 PREVIOUS HYPOTHESES

**Saliency Maps are Truthful.** Smilkov et al. (2017) suggested that noisy saliency maps are faithful descriptions of what the network is doing. That is, pixels scattered seemingly at random are actually crucial to how the network makes a decision. In short, this hypothesis claims that noise is actually informative.

**Discontinuous Gradients.** Smilkov et al. (2017) and Shrikumar et al. (2017) proposed that saliency maps are noisy due to the piece-wise linearity of the score function. Specifically, since typical DNNs use ReLU activation functions and max pooling, the derivative of the score function with respect to the input will not be continuously differentiable. Under this hypothesis, noise is caused by meaningless local variations in the gradient.

**Saturating Score Function.** Shrikumar et al. (2017) and Sundararajan et al. (2017) suggested that important features may have small gradient due to saturation. In other words, the score function can flatten in the proximity of the input and have a small derivative. This hypothesis explains why informative features may not be highlighted in the saliency map even though they contributed significantly to the decision of the DNN.

## 2.2 PREVIOUS WORKS ON IMPROVING SALIENCY MAPS

DNN interpretation methods that assign a signed *attribution* value to each input feature are collectively called *attribution methods*. Attributions are usually visualized as a heatmap by arranging them to have the same shape as the input sample. Such heatmaps are called *attribution maps*. We now describe attribution methods that have been proposed to improve saliency maps.

**Attribution Methods Addressing Discontinuity.** SmoothGrad (Smilkov et al., 2017) attempts to smooth discontinuous gradient with a Gaussian kernel. Since calculating the local average in a high dimensional space is intractable, the authors proposed a stochastic approximation which takes random samples in a neighborhood of the input $x$ and then averages their gradients.

**Attribution Methods Addressing Saturation.** Since saliency maps estimate the local importance of each input feature, they are vulnerable to saturation. Therefore, attribution methods such as Gradient * Input (Shrikumar et al., 2017), Layer-wise Relevance Propagation (LRP) (Bach et al., 2015), DeepLIFT (Shrikumar et al., 2017) and Integrated Gradient (Sundararajan et al., 2017) attempt to alleviate saturation by estimating the global importance of each pixel (Ancona et al., 2018). Ancona et al. (2018) has also shown that several global attribution methods are closely related under certain conditions.

**Other Attribution Methods.** Some attribution methods take a different approach to improving saliency maps. Deconvolution (Zeiler & Fergus, 2014) and Guided Backpropagation (Springenberg et al., 2015) remove negative gradient during backpropagation. Due to this imputation procedure, Deconvolution and Guided Backpropagation yield attribution maps sharper than those of other methods. However, Nie et al. (2018) has recently proven that these methods are actually doing partial image recovery which is unrelated to DNN decisions.

## 3 OUR EXPLANATION FOR NOISY SALIENCY MAPS

For brevity, we refer to pixels on the background as *background features* and pixels on the object as *object features*. Then, noise in a saliency map corresponds to *background gradient*, or gradient that highlights background features. We assume the DNN uses ReLU activation functions. Under this condition, nonzero background gradient indicates the presence of at least one positive pre-activation in each network layer corresponding to background features.

To verify this, we visualized intermediate layer activations of a convolutional neural network (CNN) trained on CIFAR-10. Figure 2b shows convolutional layer feature maps for an image that produced a noisy saliency map. Since CNN filters act as feature extractors, we expected the CNN to remove most background feature activations through convolutions. However, we found significant amounts of background feature activations in all convolution layers. As the last convolution layer is connected to fully connected layers, the majority of activations in the last convolution layer will have nonzero gradient. Hence, the gradient flowed through background feature activations up to the input. This gradient flow caused background gradient, as shown in Figure 2a. From our perspective, the answer to "why are saliency maps noisy?" is trivial. Saliency maps are noisy because background features pass through ReLU activation functions.

Therefore, rather than asking why saliency maps are noisy, we ask "do activations of background features highlighted by background gradient have nontrivial influence on the decision?" If the answer is *yes*, noise in saliency maps is informative as suggested by Smilkov et al. (2017), and saliency maps do not need any major improvement. However, if the answer is *no*, we should find a way to remove background gradient. We investigated this question through two experiments.

**Feature Map Occlusion.** We evaluated the significance of background features by occluding activations at intermediate layers. Then, we analyzed the effect of this perturbation on the final decision. Note that this is different from the Sensitivity metric (Bach et al., 2015; Samek et al., 2017). Sensi-

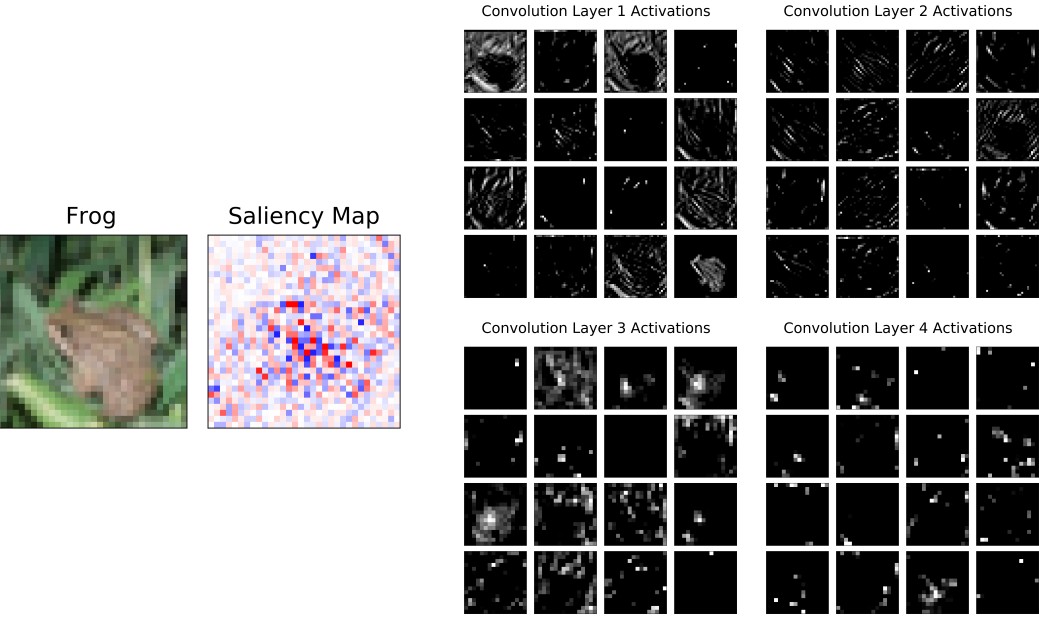

(a) Sample image and its saliency map.          (b) Intermediate layer activations.

Figure 2: Feature map visualization for an image with a noisy saliency map.

tivity measures the impact of occlusion in the data space (e.g. pixel occlusion) while we measured the impact of occlusion in each feature space.

We first created a background mask that covers background features in the images. We then plotted the average class logits as we incrementally occluded intermediate layer activations that fell on the background mask. We carried out occlusion following a random ordering and took the average over 50 trials. Figures 3a and 3b give an example of a background mask and a completely occluded feature map respectively. Figure 3c shows that the final decision did not change throughout the occlusion process for all convolution layers. Moreover, the difference between the top label logit and the next largest logit remained constant. Therefore, background feature activations are irrelevant to the classification task. To further support this claim, we conducted a larger-scale version of this experiment, and we describe the procedure and results in Appendix A.1.

**Training Dataset Occlusion.** Next, we show that gradient can be nonzero for completely uninformative features. We occluded the upper left corner of all images in the training dataset with a $10 \times 10$ random patch and trained a randomly initialized CNN on the modified dataset. We used the same patch for all images. Since the test accuracy did not change significantly ($79.4\%$ to $79.3\%$), we expected the CNN to have learned to extract important features and ignore irrelevant ones. However, Figure 4 shows that gradient is nonzero for the patch although it is completely irrelevant to the classification task.

We can draw three conclusions from these experiments:

1. DNNs do not filter out irrelevant features during forward propagation.
2. DNNs are capable of making correct decisions even if we occlude the majority of background feature activations in intermediate layers. This implies that most background feature activations are irrelevant to the classification task.
3. Since DNNs do not remove irrelevant features through ReLU activation functions, zero threshold at ReLU binary gates during backpropagation also allow irrelevant information to flow through the gradient.

With the conclusions above, we can refute the first of three previous hypotheses. As for the second hypothesis, we can interpret meaningless local variation in the gradient as a side effect of irrelevant features contaminating the gradient. Why the network does not learn to filter out irrelevant features is

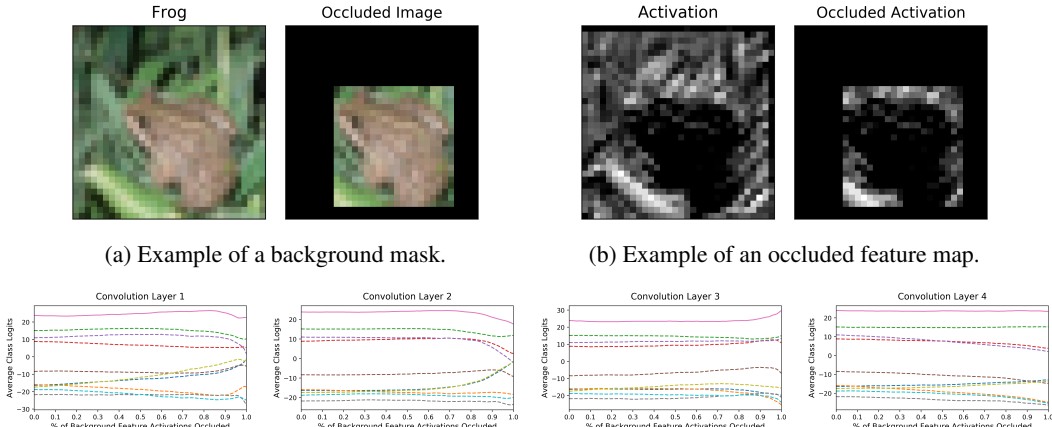

(a) Example of a background mask.

(b) Example of an occluded feature map.

(c) Average class logits as background feature activations are incrementally occluded in a random order. The average is taken over 50 trials. Image class is illustrated by a solid line and other classes by dotted lines.

Figure 3: Impact of background feature activation occlusion on the final decision.

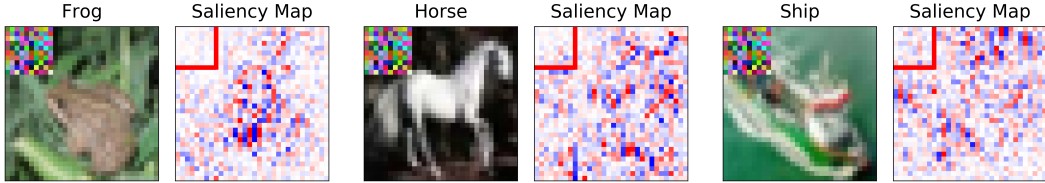

Figure 4: Saliency maps produced from a CNN trained on occluded images. The upper left corner of all the images in the training dataset is replaced with a $10 \times 10$ random patch, as shown above. Readers should examine the $8 \times 8$ patch enclosed by the red square instead of the entire $10 \times 10$ patch due to the receptive field of filters in the first convolution layer ($3 \times 3$).

a matter of optimization, which is out of scope of this paper. However, we believe it is a phenomenon worth investigating.

## 4 RECTIFIED GRADIENT

We now introduce our technique to improve saliency maps. As we have shown in Section 3, zero is a poor threshold at ReLU binary gates during backpropagation. This indicates that we need better thresholds at ReLU binary gates in order to remove uninformative gradient from saliency maps. To this end, we propose **Rectified Gradient**, or RectGrad in short, where the gradient propagates only through units whose importance scores exceed some threshold. Importance score for an unit is calculated by multiplying its activation with gradient propagated up to the unit. Formally, RectGrad is given as follows:

Suppose we have a $L$-layer ReLU DNN. Denote input feature $i$ as $x_i$, pre-activation of unit $i$ in layer $l$ as $z_i^{(l)}$, its activation as $a_i^{(l)}$ and gradient propagated up to $a_i^{(l)}$ as $R_i^{(l+1)}$. Let $\mathbb{I}(\cdot)$ be the indicator function. Then, the relation between $a_i^{(l)}$ and $z_i^{(l)}$ is given by $a_i^{(l)} = ReLU(z_i^{(l)}) = \max(z_i^{(l)}, 0)$ when $l < L$ and $a_i^{(L)} = softmax(z_i^{(L)})$. By the chain rule, backward pass through the ReLU nonlinearity for vanilla gradient is achieved by $R_i^{(l)} = \mathbb{I}(a_i^{(l)} > 0) \cdot R_i^{(l+1)}$.

We modify this rule such that $R_i^{(l)} = \mathbb{I}(a_i^{(l)} \cdot R_i^{(l+1)} > \tau) \cdot R_i^{(l+1)}$ for some threshold $\tau$. Backward pass through affine transformations and pooling operations is carried out in the same manner as backpropagation. Finally, importance scores for input features are calculated by multiplying gradient propagated up to input layer ($l = 0$) with input features: $x_i \cdot R_i^{(1)}$. Instead of setting $\tau$ to a constant

value, we use the $q^{\text{th}}$ percentile of importance scores at each layer. This prevents the gradient from entirely dying out during the backward pass.

Due to the simplicity of the propagation rule, RectGrad can easily be applied to DNNs in graph computation frameworks such as TensorFlow (Abadi et al., 2016) or PyTorch (Paszke et al., 2017). Listing 1 in Appendix D.1 shows how to implement RectGrad in TensorFlow. In Appendix C we also introduce two techniques, namely the padding trick and the proportional redistribution rule (PRR) that enhance the visual quality of RectGrad attribution maps.

### 4.1 RATIONALE BEHIND THE PROPAGATION RULE FOR RECTIFIED GRADIENT

This subsection explains the reason we have chosen $R_i^{(l)} = \mathbb{I}(a_i^{(l)} \cdot R_i^{(l+1)} > \tau) \cdot R_i^{(l+1)}$ and not $R_i^{(l)} = \mathbb{I}(a_i^{(l)} > \tau) \cdot R_i^{(l+1)}$ or $R_i^{(l)} = \mathbb{I}(R_i^{(l+1)} > \tau) \cdot R_i^{(l+1)}$ as the definition of RectGrad. The significance of multiplying an unit's activation with gradient propagated up to the unit is that it estimates the marginal effect of that unit on the output (Ancona et al., 2018). For instance, consider the following linear model: $f(a_1, a_2, a_3) = 2 \cdot a_1 + 1 \cdot a_2 + 3 \cdot a_3$. We have $\partial f / \partial a_1 = 2$, $\partial f / \partial a_2 = 1$, and $\partial f / \partial a_3 = 3$. Suppose we are given inputs $a_1 = 2$, $a_2 = 3$, $a_3 = 1$ and we apply RectGrad with $q = 67$, i.e., we propagate the gradient through the unit with the highest importance score. Clearly $a_1$ has the largest contribution of $2 \cdot 2 = 4$ to the final output compared to $1 \cdot 3 = 3 \cdot 1 = 3$ of $a_2$ and $a_3$. Only the first rule correctly propagates gradient through the most influential unit $a_1$ while the latter two rules mistakenly choose $a_2$ and $a_3$ respectively. Since the latter two rules fail even for this simple example, it is highly likely that they will not work for DNNs which are constructed by composing multiple linear layers. On the other hand, the first rule propagates gradient through units with the largest marginal effect in a layer-wise manner. Hence, it makes sense to select the first propagation rule as the definition of RectGrad. Next, we show that RectGrad generalizes Deconvolution and Guided Backpropagation.

### 4.2 RELATION TO DECONVOLUTION AND GUIDED BACKPROPAGATION

**Claim 1.** *Deconvolution * Input is equivalent to Rectified Gradient with the propagation rule*
$$R_i^{(l)} = \mathbb{I}\left[\left(a_i^{(l)} + \epsilon\right) \cdot R_i^{(l+1)} > 0\right] \cdot R_i^{(l+1)}$$
*for some small $\epsilon > 0$.*

**Claim 2.** *Guided Backpropagation * Input is equivalent to Rectified Gradient when $\tau = 0$:*
$$R_i^{(l)} = \mathbb{I}\left(a_i^{(l)} \cdot R_i^{(l+1)} > 0\right) \cdot R_i^{(l+1)}.$$

The proofs for Claims 1 and 2 are provided in Appendix E.1 and E.2 respectively. These results indicate that RectGrad generalizes Deconvolution and Guided Backpropagation. Figure 1 illustrates the relation between the saliency map, Deconvolution, Guided Backpropagation and RectGrad.

However, Nie et al. (2018) has recently proven that Deconvolution and Guided Backpropagation are actually doing partial image recovery which is unrelated to DNN decisions. RectGrad does *not* suffer from this problem as it does not satisfy the assumptions of the analyses of Nie et al. (2018) for two reasons. First, the threshold criterion is based on the product of activation and gradient which is not Gaussian distributed.[1] Second, we set $\tau$ as the $q^{\text{th}}$ percentile of importance scores and therefore $\tau$ will vary layer by layer. We also show in Section 5.2 with adversarial attacks that attributions produced by RectGrad are class sensitive. Therefore, RectGrad inherits the sharp visualizations of Deconvolution and Guided Backpropagation while amending their disadvantages with layer-wise importance score thresholding.

## 5 EXPERIMENTS

To evaluate RectGrad, we performed a series of experiments using Inception V4 network (Szegedy et al., 2017) trained on ImageNet (Russakovsky et al., 2015) and CNNs trained on CIFAR-10 (Krizhevsky & Hinton, 2009). See Appendix F.1 for details on the attribution map visualization method.

---

[1] Product of a half normal random variable and a normal random variable is not Gaussian distributed.

| Image | $\tau = 0$ | $q = 0$ | $q = 10$ | $q = 20$ | $q = 80$ | $q = 90$ | $q = 95$ | $q = 99$ |
|---|---|---|---|---|---|---|---|---|

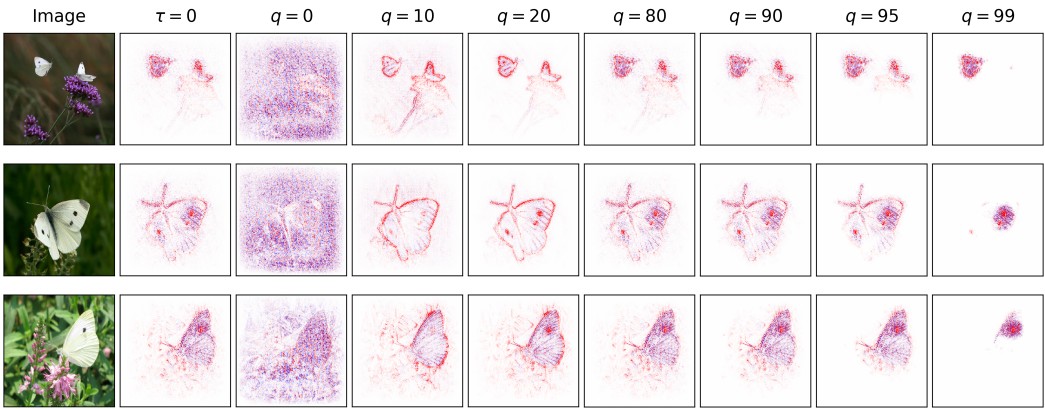

Figure 5: Effect of threshold $\tau$ (columns) on RectGrad for 3 images of the cabbage butterfly class in ImageNet (rows). The second column shows attribution maps with $\tau = 0$, which is equivalent to Guided Backpropagation * Input. For the following columns, $\tau$ is set to $q^{\text{th}}$ percentile of importance scores. The padding trick was used for all attribution maps above.

## 5.1 Effect of Threshold Percentile

RectGrad has one hyper-parameter $\tau$, which is set to $q^{\text{th}}$ percentile of importance scores for each layer. Figure 5 shows the effect of threshold percentile for several images from ImageNet. While the attribution maps were incomprehensible for $q = 0$, the visual quality dramatically improved as we incremented $q$ up to 20. There was no significant change up to $q = 80$. Then the attribution maps began to sparse out again as we incremented $q$ further. We also observed that regions of high attributions did not change from $q > 20$.

We speculate that the attributions stay constant between $q = 20$ and $80$ because of zero activations. That is, since we use ReLU activation functions, the majority of activations and consequently importance scores will be zero. Hence, $\tau \approx 0$ for $20 \leq q \leq 80$. This causes RectGrad attribution maps to resemble those produced by Guided Backpropagation * Input. It indicates that we have to increment $q > 80$ in order to produce sparser attribution maps that highlight important regions instead of reconstruct input images.

## 5.2 Qualitative Comparison with Baseline Methods

We used the saliency map, Gradient * Input, Guided Backpropagation, SmoothGrad, Integrated Gradient, Epsilon-LRP and DeepLIFT as baseline methods. As for RectGrad, we used the padding trick and $q = 98$ for all attribution maps. We show attributions both with and without application of the proportional redistribution rule. In this subsection, we compare RectGrad with other attribution methods through three experiments that each focus on different aspect of qualitative evaluation.

We also show applying simple final thresholding to baseline methods is not enough to replicate the benefits of RectGrad. To demonstrate this, we applied 95 percentile final threshold to baseline attribution methods such that RectGrad and baseline attribution maps have similar levels of sparsity.[2]

**Coherence.** Following prior work (Simonyan et al., 2014; Zeiler & Fergus, 2014), we inspected two types of visual coherence. First, the attributions should fall on discriminative features (e.g. the object of interest), not the background. Second, the attributions should highlight similar features for images of the same class.

For the first type of visual coherence, Figure 6 shows a side-by-side comparison between our method and baseline methods. It can clearly be seen that RectGrad produced attribution maps more visually coherent and focused than other methods—background noise was nearly nonexistent. This

---

[2] Note that we did not apply the threshold $q = 98$, which was used in our RectGrad results. In the setting of $q = 98$ on baseline methods, RectGrad attribution maps are slightly less sparse than baseline attribution maps. This is because threshold is applied up to the first hidden layer, not the input layer in the RectGrad procedure.

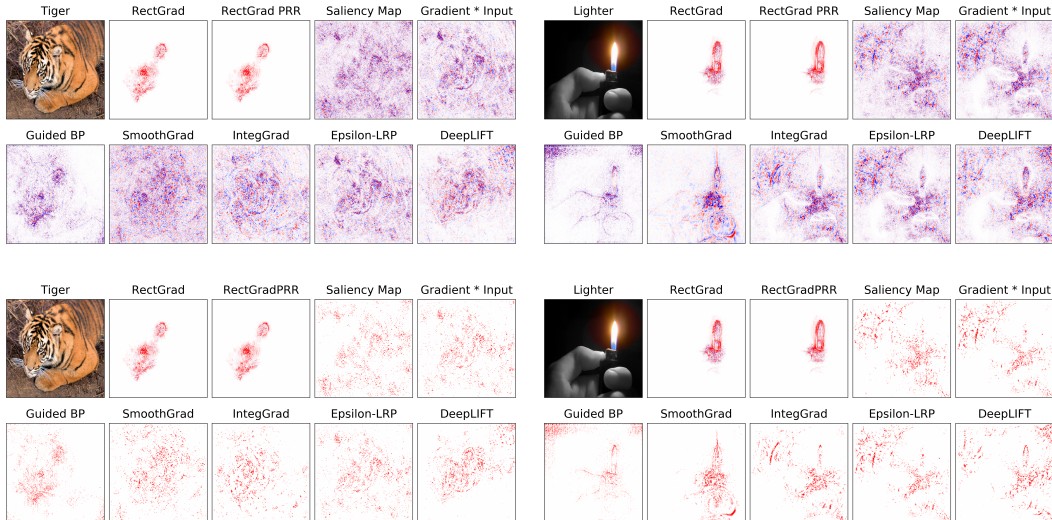

Figure 6: Evaluation of coherence across different classes without and with final thresholding.

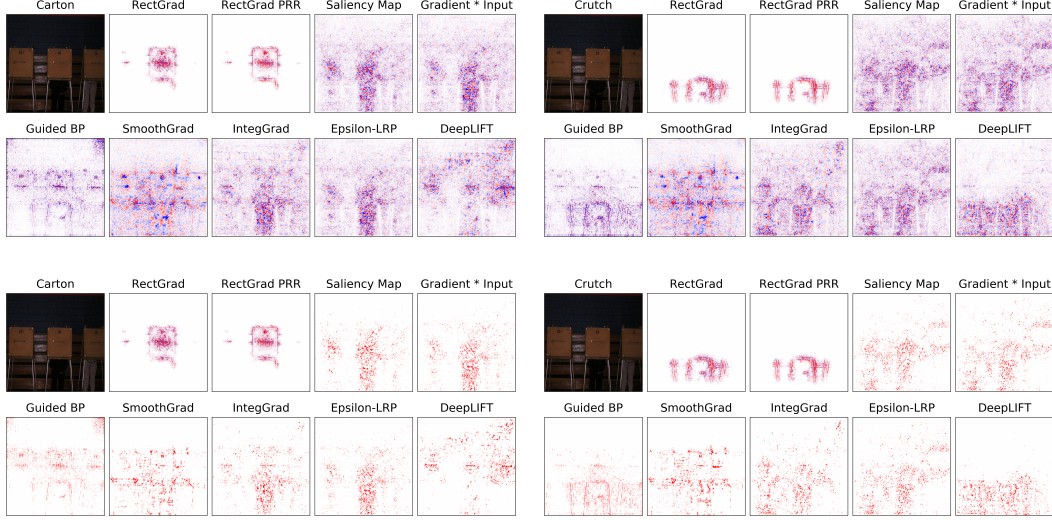

Figure 7: Comparison of attribution maps for images (left column) and their adversarial examples (right column) without and with final thresholding. This figure shows examples where attribution maps produced by RectGrad changed significantly.

phenomenon may be due to noise accumulation. Specifically, irrelevant features may have trivial gradient near the output layer. However, since gradient is calculated by successive multiplication, the noise can grow exponentially as gradient is propagated towards the input layer. This can result in confusing attribution maps which assign high attribution to irrelevant regions (e.g. uniform background in "lighter"), especially for deep networks such as Inception. RectGrad does not suffer from this problem since it thresholds irrelevant features at every layer and hence stops noise accumulation. In this situation, final thresholding cannot replicate RectGrad's ability to remove noise. In Appendix A.2, we corroborate this claim by comparing Saliency map and RectGrad attributions as they are propagated towards the input layer.

For the second type of visual coherence, Figure 11 in Appendix A.3 shows attribution maps for a pair of images belonging to the same class. Attribution maps generated by RectGrad consistently emphasized similar parts of the object of interest. On the contrary, Saliency map, Gradient * Input and Epsilon-LRP emphasized different regions for each image instance. Attributions for Smooth-

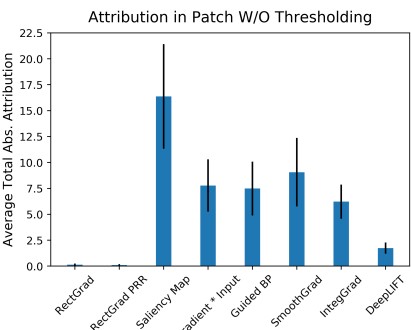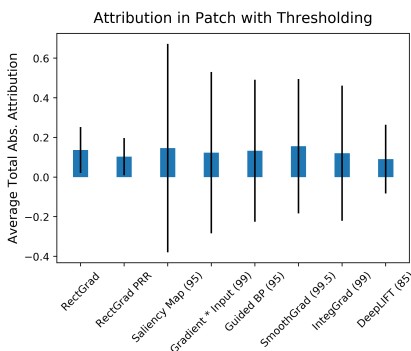

Figure 8: Comparison of amount of attribution on occluded patch. The left and right charts compare the amount of attribution inside occluded patch without and with final thresholding respectively. The numbers in parentheses show the custom threshold levels.

Grad, Guided Backpropagation, Integrated Gradient and DeepLIFT were generally coherent across images of the same class. Nevertheless, they also highlighted background features and hence failed to satisfy the first type of visual coherence. This observation also holds for attribution maps with final thresholding.

**Adversarial Attack.** We evaluated class sensitivity following prior work by Nie et al. (2018). Specifically, we compared the attributions for an image and its adversarial example. If the attribution method is class sensitive, attribution maps should change significantly since ReLU activations and consequently the predicted class have changed. On the other hand, if the attribution method merely does image reconstruction, attribution maps will not change much since we add an indistinguishable adversarial perturbation to the image. In this experiment, we used the fast gradient sign method (Goodfellow et al., 2015) with $\epsilon = 0.01$ to generate adversarial examples.

Figure 7 shows large changes in attribution maps produced by RectGrad. We observed that only RectGrad attributions were coherent with the class labels. Figure 12 in Appendix A.3 shows some instances where there was no significant change in attribution maps produced by RectGrad. In those cases, attribution maps for other methods also showed little change. Hence, we can conclude that RectGrad is equally or more class sensitive than baseline attribution methods. We observed that this conclusion also holds with final thresholding. It is also possible that adversarial attacks only modified a tiny amount of ReLU activations (i.e. the images were near the decision boundary), causing little change in attribution maps.

### 5.3 QUANTITATIVE COMPARISON WITH BASELINE METHODS

In this section, we quantitatively compare RectGrad with baseline methods using DNNs trained on CIFAR-10. We did not include Epsilon-LRP since it is equivalent to Gradient * Input for ReLU DNNs (Ancona et al., 2018). We divided baseline attribution methods into local and global methods following the criterion proposed by Ancona et al. (2018). We also repeated the same experiments with final thresholding to the baselines to compare them with RectGrad in similar sparsity setting.

**Training Dataset Occlusion.** Just like the training dataset occlusion experiment in Section 3, we occluded the upper left corner of all images in CIFAR-10 training dataset with a $10 \times 10$ random patch and trained a randomly initialized CNN on the modified dataset. We then summed all absolute attribution within the patch and averaged across the test dataset. A reasonable attribution method should assign nearly zero attribution to the patch as it is completely irrelevant to the classification task. Figure 8 compares the amount of average attribution in the patch between attribution methods. We observed that without final thresholding, RectGrad assigned little or no attribution to the random patch. However, all other methods failed to do so. For this test, we found using $q = 95$ final threshold led to trivially different averages. Hence we used a custom threshold for each baseline method such that they had similar average attribution in the patch as RectGrad. We observed that RectGrad had smaller standard deviation than baseline methods. This indicates that RectGrad more consistently assigns near-zero attribution to the patch. Therefore RectGrad has advantages over

baseline methods regardless of whether final threshold is used or not. Figures for the following quantitative experiment outcomes are in Appendix A.4.

**Noise Level.** We evaluated whether RectGrad really reduces noise through two experiments. For the first test, we created segmentation masks for 10 correctly classified images of each class (total 100 images) and measured how much attribution falls on the background. Specifically, we compared the sum of absolute value of attribution on the background. For the second test, we measured the average total variation of attribution maps for each attribution method. The average was taken over the test dataset. Figure 13 shows that RectGrad assigned significantly less attribution to the background than baseline methods. Moreover, even with final thresholding, RectGrad outperformed baseline methods. In addition, Figure 14 shows that even though the total variation reduces for baseline methods after final thresholding, RectGrad outperforms baseline methods in both cases. The results imply that baselines with final thresholding cannot replicate RectGrad's ability to reduce noise.

**Sensitivity.** We evaluated RectGrad using the Sensitivity metric proposed by Bach et al. (2015) and Samek et al. (2017). Specifically, we measured how the logit for the initial class changed as features were occluded based on the ordering assigned by the attribution method. We split the image into non-overlapping patches of $2 \times 2$ pixels. Next, we computed attributions and summed all the values within each patch. We sorted the patches in decreasing order based on the aggregate attribution values. We then incrementally replaced the first 100 patches with per-channel mean computed using the entire training set and measured the change in class logit. We calculated the average across 500 randomly chosen test set images. An attribution method is better if it has a lower sensitivity AUC.

The results are shown in Figure 15. All attribution methods outperformed the random baseline in which we randomly removed patches. We observed that RectGrad performed better than local attribution methods. In comparison with global attribution methods, RectGrad showed similar performance up to approximately 10 patches (red vertical line) but the performance dropped as more patches were removed. In Appendix B.1, we offer an explanation for this behavior. Figure 16 shows that after final thresholding, RectGrad still outperforms local attribution methods. For global attribution methods, RectGrad now shows similar performance.

**ROAR and KAR.** We evaluated RectGrad using Remove and Retrain (ROAR) and Keep and Retrain (KAR) proposed by Hooker et al. (2018). Specifically, we measured how the performance of the classifier changed as features were occluded based on the ordering assigned by the attribution method. For ROAR, given an attribution method, we replaced a fraction of all CIFAR-10 pixels that were estimated to be *most* important with a constant value. We then retrained a CNN on the modified dataset and measured the change in test accuracy. For KAR, we replaced a fraction of all CIFAR-10 pixels that were estimated to be *least* important. We trained 3 CNNs per estimator for each fraction $\{0.1, 0.3, 0.5, 0.7, 0.9\}$. We measured test accuracy as the average of theses 3 CNNs. An attribution method is better if it has a lower ROAR AUC and a higher KAR AUC.

Figure 17 presents ROAR scores. All attribution methods outperformed the random baseline in which we randomly removed pixels. RectGrad showed similar performance to local attribution methods but performed worse than all global attribution methods. Next, Figure 18 shows KAR scores. Interestingly, all baseline attribution methods failed to exceed even the random baseline. Only RectGrad had similar or better performance than the random baseline. In Appendix B.2, we offer an explanation for why RectGrad performed poorly in ROAR.

## 6 CONCLUSIONS

Saliency map is the most basic means of interpreting deep neural network decisions. However, it is often visually noisy. Although several hypotheses were proposed to account for this phenomenon, there is no work that provides a thorough analysis of noisy saliency maps. Therefore, we first identified saliency maps are noisy because DNNs do not filter out irrelevant features during forward propagation. We then proposed RectGrad Gradient which significantly improves saliency maps by alleviating this problem through layer-wise thresholding during backpropagation. We showed that Rectified Gradient generalizes Deconvolution and Guided Backpropagation and moreover, overcomes the class-insensitivity problem. We also demonstrated through extensive experiments that Rectified Gradient outperforms previous attribution methods.

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

# A EXPERIMENT RESULTS

## A.1 SUPPLEMENTARY EXPERIMENT FOR FEATURE MAP OCCLUSION

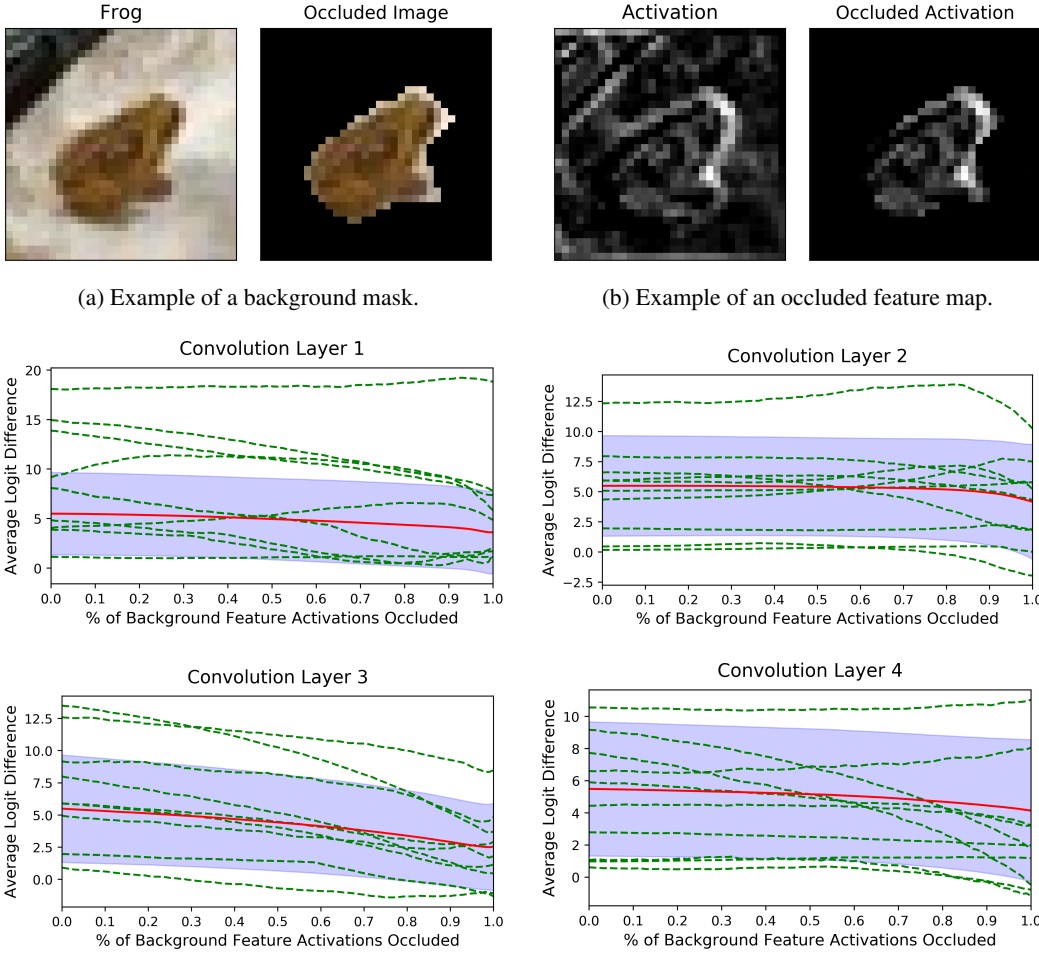

(a) Example of a background mask.     (b) Example of an occluded feature map.

(c) Average of (class logit) − (largest logit among the other 9 classes) as background feature activations are incrementally occluded in a random order (average is taken over 50 random trials). The average is taken over 100 images. The average is illustrated by a solid red line, standard deviation by the shaded blue region, and 10 randomly selected instances by green dotted lines.

Figure 9: Larger-scale study of the impact of background feature activation occlusion on the final decision.

To further support our claim that background feature activations are irrelevant to the classification task, we conducted a larger-scale experiment. We created segmentation masks for 10 correctly classified images of each class (total 100 images) and repeated the feature map occlusion for each image. We then took the average of (class logit) − (largest logit among the other 9 classes) across all 100 images. Figures 9a and 9b give an example of a background segmentation mask and a completely occluded feature map. Figure 9c shows that the difference is generally positive throughout the occlusion process, that is, the class does not change for most images. From this, we can infer that background features are generally irrelevant to the classification task.

## A.2    SUPPLEMENTARY EXPERIMENT FOR NOISE ACCUMULATION

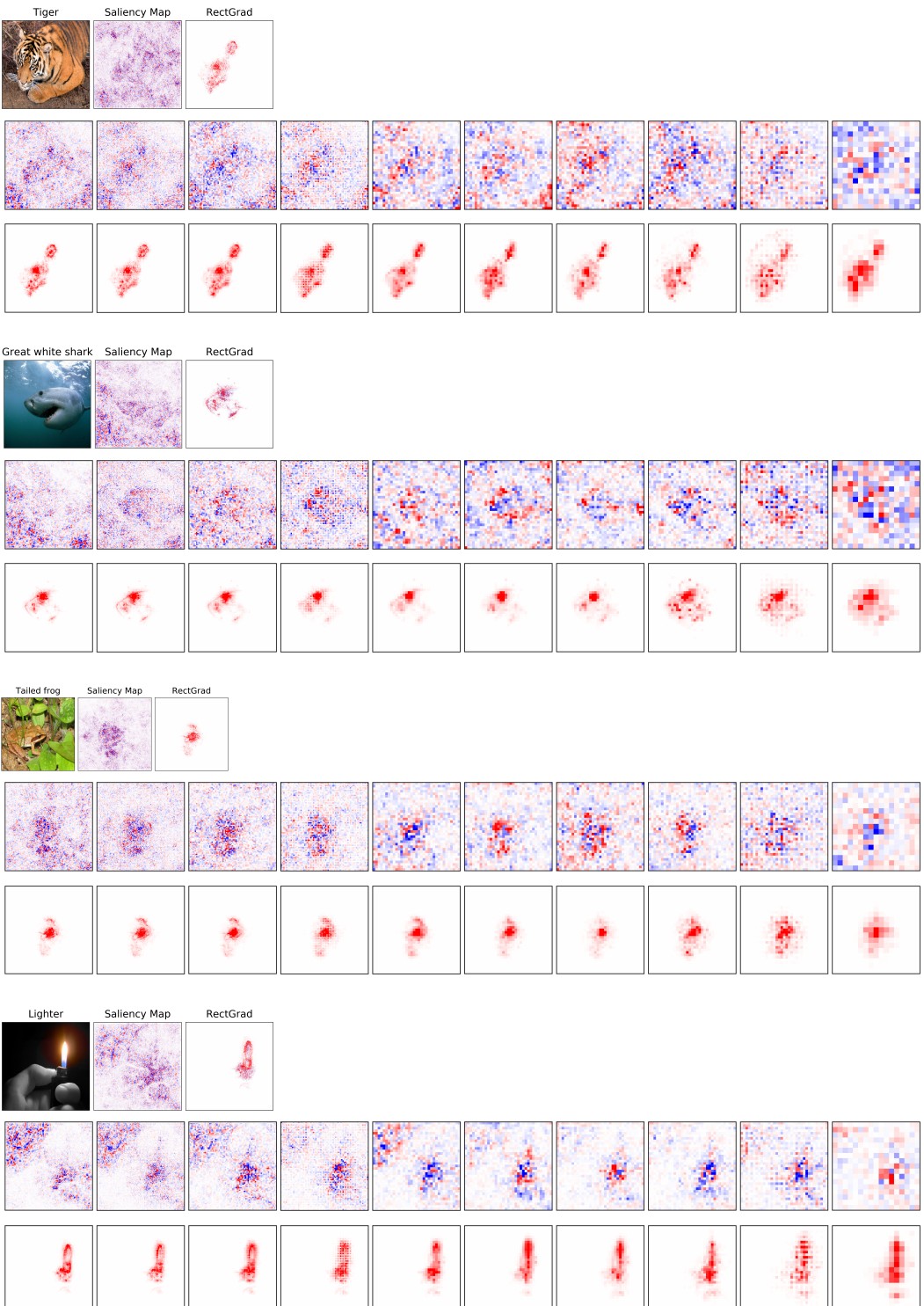

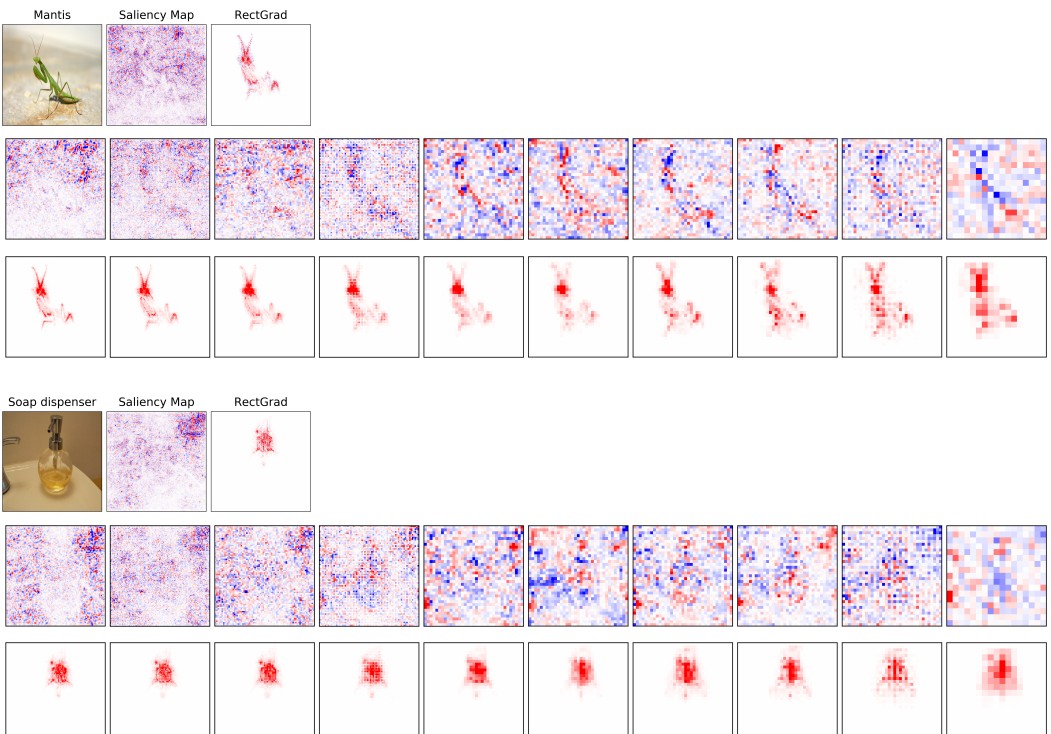

Figure 10: Saliency map and RectGrad attributions at Inception v4 intermediate layers as they are propagated toward the input layer. We show channel-wise average attributions for hidden layer inputs with respect to the output layer. For each subfigure, first row shows the input image and Saliency map and RectGrad attribution maps. Second and third rows show Saliency map and Rect-Grad attributions at intermediate layers, respectively. An attribution map is closer to the output layer if it is closer to the right.

To verify our claims on the noise accumulation phenomenon, we compared Saliency map and Rect-Grad attributions as they are propagated towards the input layer. As Figure 10 shows, at higher layers, Saliency map attributions for objects of interest are generally larger than or equal to attributions on the background. However, as they are propagated towards the input layer, attributions for objects of interest diminish while background attributions grow. On the other hand, RectGrad removes background attributions from higher layers through importance score based thresholding, stopping noise accumulation in the first place.

## A.3 QUALITATIVE EXPERIMENTS

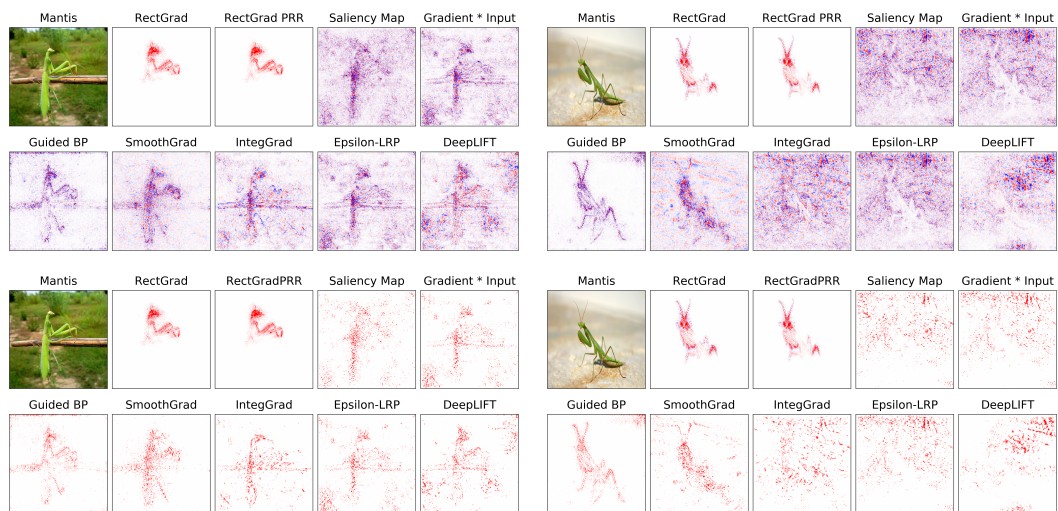

Figure 11: Evaluation of coherence within the same class (rows) without and with final thresholding.

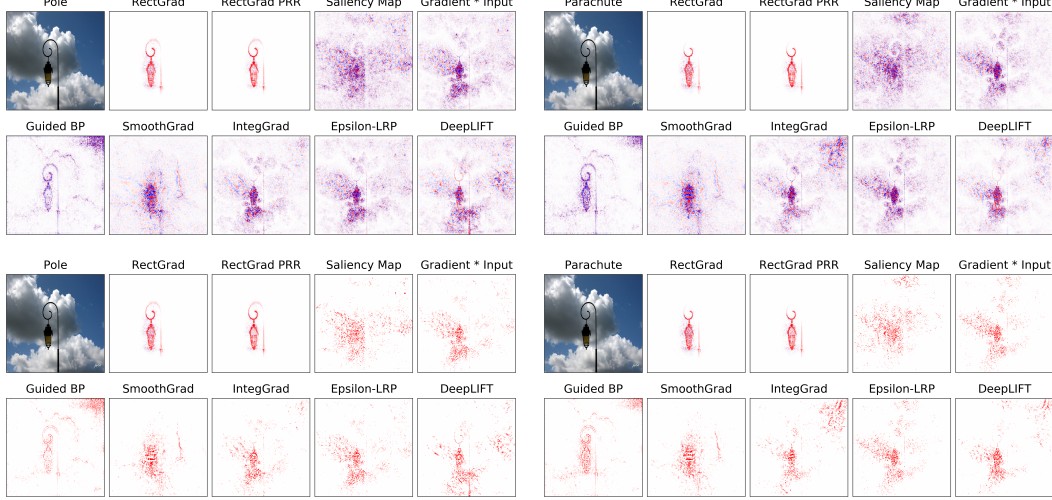

Figure 12: Comparison of attribution maps for images (left column) and their adversarial examples (right column) without and with final thresholding. This figure shows examples where attribution maps produced by RectGrad did not change significantly.

## A.4 QUANTITATIVE EXPERIMENTS

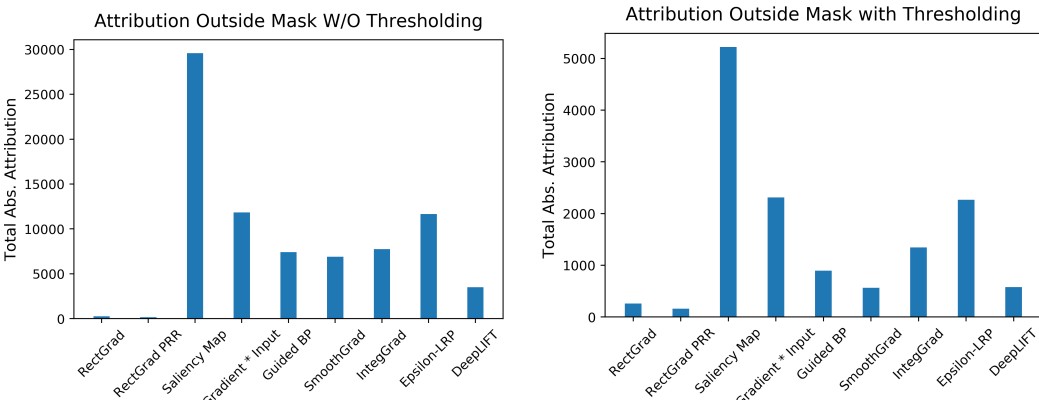

Figure 13: Comparison of amount of attribution on the background. The left and right charts compare the amount of attribution outside mask (on background) without and with final thresholding respectively.

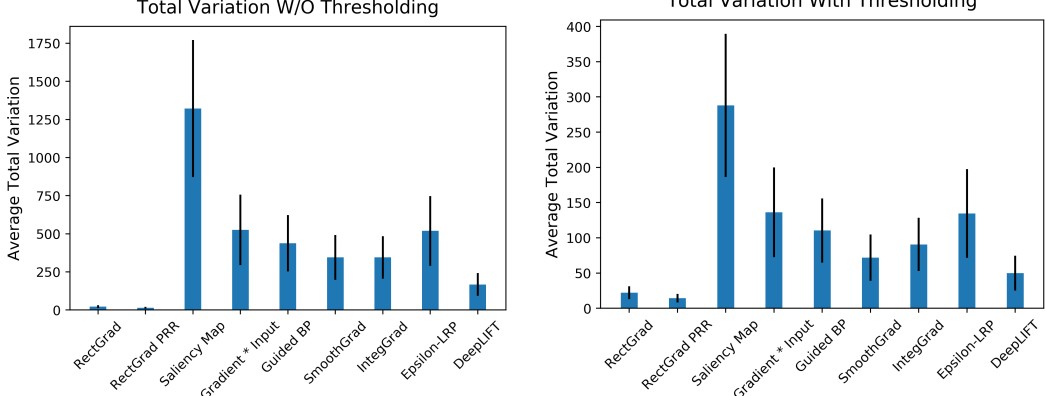

Figure 14: Comparison of average total variation. The left and right charts compare average total variation without and with final thresholding respectively.

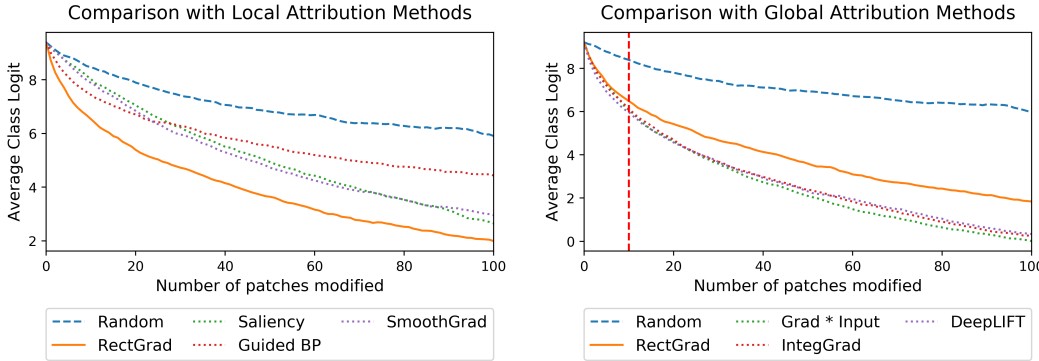

Figure 15: Comparison of Sensitivity. The left plot compares RectGrad with local attribution methods and the right plot with with global attribution methods. We also include the random baseline (patches are randomly removed) for reference. Lower AUC indicates a better attribution method. The red vertical line in the right plot indicates where RectGrad starts to perform worse than baseline global attribution methods (10 patches). We took the average over 500 randomly chosen test set images.

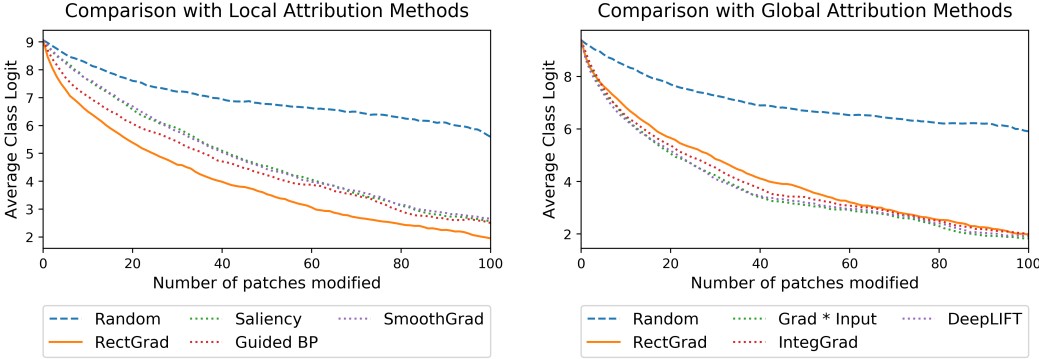

Figure 16: Comparison of Sensitivity after final thresholding. The left plot compares RectGrad with local attribution methods and the right plot with with global attribution methods. We also include the random baseline (patches are randomly removed) for reference. Lower AUC indicates a better attribution method. We took the average over 500 randomly chosen test set images.

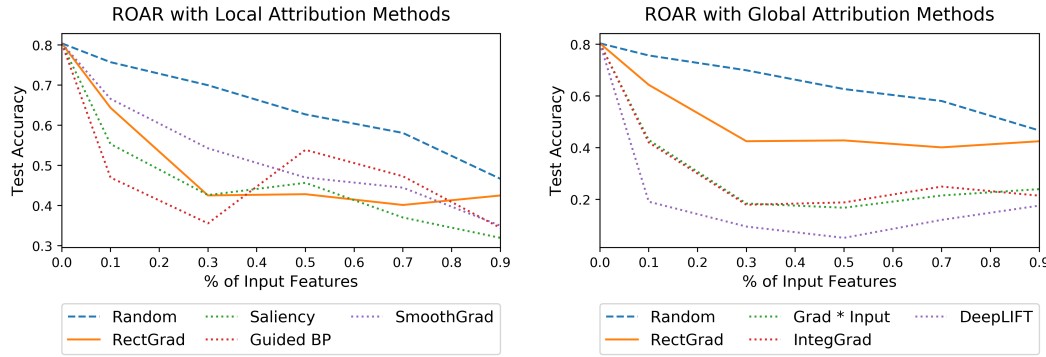

Figure 17: Comparison of ROAR. The left plot compares RectGrad with local attribution methods and the right plot with with global attribution methods. We also include the random baseline (pixels are randomly removed) for reference. Lower AUC indicates a better attribution method.

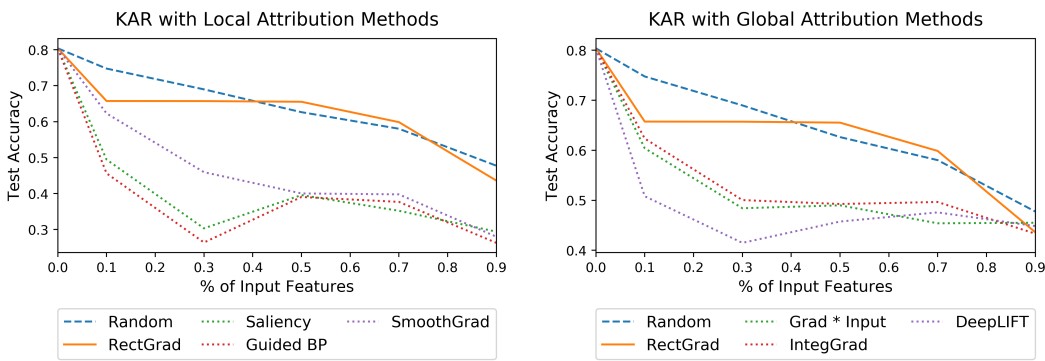

Figure 18: Comparison of KAR. The left plot compares RectGrad with local attribution methods and the right plot with with global attribution methods. We also include the random baseline (pixels are randomly removed) for reference. Higher AUC indicates a better attribution method.

# B  ADDITIONAL EXPLANATION FOR QUANTITATIVE EXPERIMENTS

## B.1  SENSITIVITY

In comparison with global attribution methods, RectGrad showed similar performance up to approximately 10 patches (red vertical line) but the performance dropped as more patches were removed (Figure 15). We speculate this happens due to the sparseness of RectGrad attribution maps. Since RectGrad attribution maps are sparser than those of other methods, occluding approximately 10 features will be enough to remove core features highlighted by RectGrad. Attributions for other features will not be as informative since they have trivial values. Figure 19 shows that it is indeed the case. For RectGrad, after occluding 10 top $2 \times 2$ patches, only attributions of small values remained. For Gradient * Input, on the other hand, still had significant amount of nontrivial leftover attributions. We also see from Figure 16 that this phenomenon also happens for baseline methods with final thresholding. This implies that such behavior may be an inevitable consequence of sparseness.

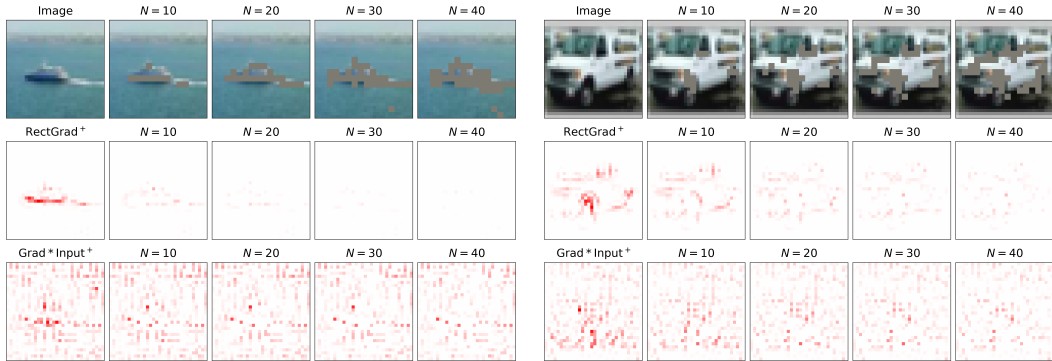

Figure 19: Comparison of attribution methods in sensitivity. The first row shows the image as top $N$ $2 \times 2$ patches are occluded according to RectGrad. The second and third rows show the positive parts (indicated by $+$) of RectGrad and Gradient * Input attribution maps as top $N$ $2 \times 2$ patches are occluded respectively. We did not cap outlying values in this visualization.

### B.2 ROAR

We believe that the poor performance of RectGrad in ROAR is also due to its sparseness. Since RectGrad produces visually coherent attribution maps, the occluded regions can act as discriminative features. To verify this, we replaced 10% of all CIFAR-10 pixels that were estimated to be most important with the channel-wise mean. We then trained a CNN on the occluded dataset and visualized RectGrad attribution maps for images whose original and occluded versions were both classified correctly. Figure 20 shows the results. Attribution maps highlighted pixels around the occluded regions and moreover, similar regions were emphasized in the original image. This corroborates our claim that the occluded regions act as discriminative features. The assumption behind ROAR is that the occluded features do not influence the classification task (Hooker et al., 2018). Since the above observation contradicts this assumption, ROAR may not be suitable for objectively evaluating RectGrad.

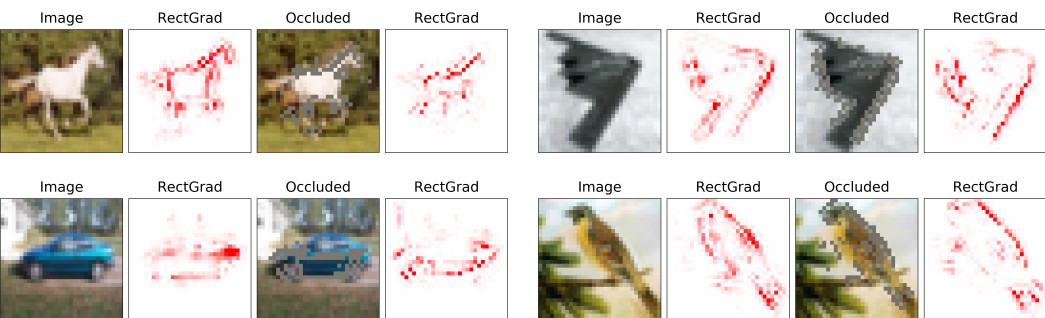

Figure 20: RectGrad attribution maps produced from a CNN trained on images occluded according to RectGrad. We show images whose original and occluded versions were both classified correctly.

## C USEFUL TECHNIQUES

Here, we present two useful techniques that can enhance the visual quality of attribution maps produced by RectGrad.

### C.1 PADDING TRICK

Convolution inputs are typically zero padded along the border in order to preserve the spatial dimension of feature maps.[3] This occasionally leads to high activation values along the border if zero is out of input distribution. Since importance scores are calculated by multiplying activation with gradient, outlying border activation can cause RectGrad to be propagated through the border instead of relevant features. To solve this problem, we masked the border of gradient to zero before the backward pass through convolutions with padding. One possible concern with the padding trick is that attributions may be faint for features adjacent to the border of the image. However, we did not find this to be a significantly problem experimentally. Listing 2 in Appendix D.2 shows how to implement the padding trick in TensorFlow.

### C.2 PROPORTIONAL REDISTRIBUTION RULE (PRR) FOR POOLING LAYERS.

Attribution maps produced by RectGrad tend to be rough due to the discrete nature of thresholding. This discontinuity can be compensated by using the proportional redistribution rule proposed by Montavon et al. (2017) for the backward pass through max-pooling layers. Instead of propagating the gradient through only the most activated unit in the pool, gradient is redistributed proportional to unit activations. Since the redistribution operation is continuous, attribution maps generated with the proportional redistribution rule are smoother. Listing 3 in Appendix D.3 shows how to implement the proportional redistribution rule in TensorFlow.

---

[3]This corresponds to convolution with SAME padding in TensorFlow terminology.

## D  TENSORFLOW CODES

### D.1  IMPLEMENTATION OF RECTIFIED GRADIENT

```python
import tensorflow as tf

from tensorflow.contrib.distributions import percentile

@tf.RegisterGradient("RectifiedRelu")
def _RectifiedReluGrad(op, grad):

    def threshold(x, q):

        if len(x.shape.as_list()) > 3:
            thresh = percentile(x, q, axis=[1,2,3], keep_dims=True)
        else:
            thresh = percentile(x, q, axis=1, keep_dims=True)

        return thresh

    activation_grad = op.outputs[0] * grad
    thresh = threshold(activation_grad, q)

    return tf.where(thresh < activation_grad, grad, tf.zeros_like(grad))
```

Listing 1: Implementation of Rectified Gradient in TensorFlow. After registering this function as the gradient for ReLU activation functions, call `tf.gradients()` and multiply with inputs to generate attributions.

### D.2  IMPLEMENTATION OF THE PADDING TRICK

```python
import tensorflow as tf

@tf.RegisterGradient("RectifiedConv2D")
def _RectifiedConv2DGrad(op, grad):

    if op.get_attr('padding') == b'SAME':

        shape = tf.shape(grad)
        mask = tf.ones([shape[0], shape[1] - 2, shape[2] - 2, shape[3]])
        mask = tf.pad(mask, [[0,0],[1,1],[1,1],[0,0]])
        grad = grad * mask

    input_grad = tf.nn.conv2d_backprop_input(tf.shape(op.inputs[0]), op.inputs[1], grad, op.get_attr('strides'), op.get_attr('padding'))
    filter_grad = tf.nn.conv2d_backprop_filter(op.inputs[0], tf.shape(op.inputs[1]), grad, op.get_attr('strides'), op.get_attr('padding'))

    return input_grad, filter_grad
```

Listing 2: Implementation of the padding trick in TensorFlow. After registering this function as the gradient for convolution operations, call `tf.gradients()` and multiply with inputs to generate attributions.

### D.3    IMPLEMENTATION OF THE PROPORTIONAL REDISTRIBUTION RULE

```python
import tensorflow as tf

from tensorflow.python.ops import gen_nn_ops

@tf.RegisterGradient("RectifiedMaxPool")
def _RectifiedMaxPoolGrad(op, grad):

    z = tf.nn.avg_pool(op.inputs[0], op.get_attr('ksize'), op.get_attr('
    strides'), op.get_attr('padding')) + 1e-10
    s = grad / z
    c = gen_nn_ops._avg_pool_grad(tf.shape(op.inputs[0]), s, op.get_attr(
    'ksize'), op.get_attr('strides'), op.get_attr('padding'))

    return op.inputs[0] * c
```

Listing 3: Implementation of the proportional redistribution rule in TensorFlow. After registering this function as the gradient for max-pooling operations, call `tf.gradients()` and multiply with inputs to generate attributions.

## E    PROOF OF CLAIMS

### E.1    PROOF OF CLAIM 1

*Proof.* Note that the backward propagation rule for Deconvolution through the ReLU nonlinearity is given by

$$R_i^{(l)} = \mathbb{I}\left(R_i^{(l+1)} > 0\right) \cdot R_i^{(l+1)}. \tag{1}$$

Since the DNN uses ReLU activation functions, $a_i^{(l)} + \epsilon > 0$ and therefore

$$\mathbb{I}\left[\left(a_i^{(l)} + \epsilon\right) \cdot R_i^{(l+1)} > 0\right] = \mathbb{I}\left(R_i^{(l+1)} > 0\right) \tag{2}$$

for all $l$ and $i$. The result follows from Equation 2.    □

### E.2    PROOF OF CLAIM 2

*Proof.* Note that the backward propagation rule for Guided Backpropagation through the ReLU nonlinearity is given by

$$R_i^{(l)} = \mathbb{I}\left(z_i^{(l)} > 0\right) \cdot \mathbb{I}\left(R_i^{(l+1)} > 0\right) \cdot R_i^{(l+1)}. \tag{3}$$

Since the DNN uses ReLU activation functions, $a_i^{(l)} \geq 0$ and therefore

$$\mathbb{I}\left(a_i^{(l)} \cdot R_i^{(l+1)} > 0\right) = \mathbb{I}\left(z_i^{(l)} > 0\right) \cdot \mathbb{I}\left(R_i^{(l+1)} > 0\right) \tag{4}$$

for all $l$ and $i$. The result follows from Equation 4.    □

## F    EXPERIMENTS SETUP

### F.1    ATTRIBUTION MAP VISUALIZATION

To visualize the attributions, we summed up the attributions along the color channel and then capped low outlying values to $0.5^{\text{th}}$ percentile and high outlying values to $99.5^{\text{th}}$ percentile for RGB images. We only capped outlying values for grayscale images.

### F.2    CIFAR-10

The CIFAR-10 dataset (Krizhevsky & Hinton, 2009) was pre-processed to normalize the input images into range $[-1; 1]$. We trained a CNN using ReLU activation functions with Adam for 20 epochs to achieve $79.4\%$ test accuracy. For the dataset occluded with the random patch, we used the same settings to achieve $79.3\%$ test accuracy.

| CIFAR-10 CNN |
|:---:|
| Conv 2D ($3 \times 3$, 32 kernels) |
| Conv 2D ($3 \times 3$, 32 kernels) |
| Max-pooling ($2 \times 2$) |
| Dropout (0.25) |
| Conv 2D ($3 \times 3$, 64 kernels) |
| Conv 2D ($3 \times 3$, 64 kernels) |
| Max-pooling ($2 \times 2$) |
| Dropout (0.25) |
| Dense (256) |
| Dropout (0.5) |
| Dense (10) |

### F.3    INCEPTION V4

We used a pre-trained Inception V4 network. The details of this architecture can be found in Szegedy et al. (2017). For the adversarial attack, we used the fast gradient sign method with $\epsilon = 0.01$.

