# OpenReview forum: "Rectified Gradient: Layer-wise Thresholding for Sharp and Coherent Attribution Maps"
_ICLR.cc/2019/Conference_

### Official Review · AnonReviewer3 · 2018-10-24
**Insightful observations, but results are less convincing.**

**Rating:** 4
**Confidence:** 4

**Review:**

In the paper, the authors proposed a new saliency map method, based on some empirical observations about the cause of noisy gradients.
Specifically, through experiments, the authors clarified that the noisy gradients are due to irrelevant information propagated in the forward pass in DNN. Because the backpropagation follows the same pass, irrelevant feature are conveyed back to the input, which results in noisy gradients.
To avoid noisy gradients, the authors proposed a new backpropagation named Rectified Gradient (RectGrad). In RectGrad, the backward pass is filtered out if the product of the forward signal and the backward signal are smaller than a threshold. The authors claim that, with this modification in backpropagation, the gradients get less noisy.
In some experiments, the authors presented that RectGrad can produce clear saliency maps.

I liked the first half of the paper: the observations that irrelevant forward passes are causing noisy gradients seem to be convincing. The experiments are designed well to support the claim.
Here, I would like to point out, that noisy gradients in occluded images may be because of the convolutional structures. Each filter in convolution layer is trained to respond to certain patterns. Because the same filter is used for each of subimages, some filters can be activated occasionally on occluded parts. I think this does not happen if the network is densely connected without convolutional structures. The trained dense connection will be optimized to remove the effects of occluded parts. Hence, for such networks, the gradient will be zeros for occluded parts.

The second half of the paper (Sec.4 and 5) are not very much convincing to me.
Below, I raise several concerns.

1. There is no justification on the definition of RectGrad: Why Rl = I(al * Rl > t) R(l+1)?
The authors presented Rl = I(al * Rl > t) R(l+1) as RectGrad, that can filter out irrelevant passes. However, there is no clear derivation of this formula: the definition suddenly appears. If the irrelevant forward passes are causes of noisy gradients, the modification Rl = I(al > t) R(l+1) seems to be more natural to me. It is also a natural extension to the ReLU backward pass Rl = I(al > 0) R(l+1). Why we need to filter out negative signals in backward pass?

2. The experimental results are less convincing: Is RectGrad truly good?
In Sec.5.2, the authors presented saliency maps only on a few images, and claimed that they look nicely. However, it is not clear that those "nicely looking" saliency map are truly good ones. I expect the authors to put much efforts on quantitative comparisons rather than qualitative comparisons, so that we can understand that those "nicely looking" saliency maps are truly good ones.
Sec.5.3 presents some quantitative comparisons, however, the reported Sensitivity and ROAR/KAR on RectGrad are not significant. The authors mentioned that this may be because of the sparsity of RectGrad. However, if the sparsity is the harm, the underlying observations of RectGrad may have some errors. I think the current manuscript has an inconsistency between the fundamental idea (based on empirical observations) and the performance of RectGrad.

[Minor Concern]
In Sec.5, the authors frequently refer to the figures in appendix. I think the main body of the paper should be self-contatined. I therefore think that some of the figures related to main results should appear in the main part.

---

> ### Author Response · Authors · 2018-11-16
> **Reply to AnonReviewer3 Part 1/3**
>
> Thank you for the friendly and detailed review. Before reading our reply for your review, we politely ask you to read “Our Common Reply to All Reviewers” first.
>
> C1: "I liked the first half of the paper: the observations that irrelevant forward passes are causing noisy gradients seem to be convincing. The experiments are designed well to support the claim. Here, I would like to point out, that noisy gradients in occluded images may be because of the convolutional structures. Each filter in convolution layer is trained to respond to certain patterns. Because the same filter is used for each of subimages, some filters can be activated occasionally on occluded parts. I think this does not happen if the network is densely connected without convolutional structures. The trained dense connection will be optimized to remove the effects of occluded parts. Hence, for such networks, the gradient will be zeros for occluded parts."
>
> A1: Thank you for the compliments! Note that if we only use dense layers as the Reviewer suggested, we would not be able to achieve such high test accuracy. Hence the problem of noisy gradients for CNNs  would have to be addressed sooner or later. In addition, at the preliminary stage of this research, we also found that using fully connected layers does not solve this problem. We speculate this happens due to two reasons: (1) random initialization of weights and (2) lack of incentive for the network to "not remove forward signal from irrelevant features" or "zero out weights corresponding to irrelevant features". Correspondingly, we found that using l2 loss / weight decay k||w||^2 does remove noise from saliency maps. However, by the time the weight decay coefficient k was high enough to produce clear saliency maps, DNN lacked the expressiveness to achieve sufficiently high test accuracy. Therefore, we did not include this observation in the final version of our paper.
>
> C2: "There is no justification on the definition of RectGrad: Why Rl = I(al * Rl > t) R(l+1)? The authors presented Rl = I(al * Rl > t) R(l+1) as RectGrad, that can filter out irrelevant passes. However, there is no clear derivation of this formula: the definition suddenly appears. If the irrelevant forward passes are causes of noisy gradients, the modification Rl = I(al > t) R(l+1) seems to be more natural to me. It is also a natural extension to the ReLU backward pass Rl = I(al > 0) R(l+1). Why we need to filter out negative signals in backward pass?"
>
> A2: We direct the Reviewer to Section 4.1 where we explain the rationale behind the definition of RectGrad in the revised version of the paper. We also direct the Reviewer to A2 of our reply to AN2, where we answer "how does RectGrad compare with simply applying a final threshold on other attribution maps?".
>
> C3: "The experimental results are less convincing: Is RectGrad truly good? In Sec.5.2, the authors presented saliency maps only on a few images, and claimed that they look nicely.”
>
> A3: We direct the Reviewer to A1 of our reply to AN1, where we reply to "One of my biggest concern is regarding the experiment and evaluation section. Conclusions are drawn based on the visualization of a few saliency maps. I am not sure how much I can trust these conclusions as the conclusions are drawn in a handy-wavy manner the examples are prone to cherry-picking.”
>
> We have also proven through additional quantitative experiments that RectGrad attribution maps are not only sparse, but significantly less noisy than baseline attribution maps. We direct the Reviewer to A5 of our reply to AN1, where we answer "since this paper claims to produce less noisy saliency maps, what does it mean quantitatively? Is it true that it produces less pixels on the background? If so, can we evaluate it with foreground-background segmentation annotation to prove that point?"

---

> > ### Author Response · Authors · 2018-11-16
> > **Reply to AnonReviewer3 Part 2/3**
> >
> > C4: “However, it is not clear that those "nicely looking" saliency map are truly good ones. I expect the authors to put much efforts on quantitative comparisons rather than qualitative comparisons, so that we can understand that those "nicely looking" saliency maps are truly good ones."
> >
> > A4: The concern seems to be about whether the sparse attribution maps produced by RectGrad are truly meaningful. If this is not the case, then RectGrad should perform worse than baseline methods with final threshold. That is, RectGrad attribution maps should have no advantage over baseline attribution maps that are simply thresholded to have the same level of sparsity. To verify whether this is true, we applied final threshold so that baseline attribution maps have similar sparsity as RectGrad attribution maps and then repeated all the experiments. We also conducted additional quantitative experiments to support our claims.
> >
> > The experimental settings and results are described in A3 of our reply to AN2, where we answer "how do the results on training data and feature occlusion change after such a threshold is applied? How do results on adversarial attacks change?". We also direct the Reviewer to A3 of our reply to AN1, where we reply to "it is stretching to conclude that it is more class sensitive without further quantitative validation."
> >
> > C5: "Sec.5.3 presents some quantitative comparisons, however, the reported Sensitivity and ROAR/KAR on RectGrad are not significant."
> >
> > A5: We would first like to point out that the result for KAR is not trivial. On the contrary, RectGrad shows the best performance among all attribution methods on KAR. We cite our description of the results on KAR in the paper: “next, Figure 18 shows KAR scores. Interestingly, all baseline attribution methods failed to exceed even the random baseline. Only Rectiﬁed Gradient had similar or better performance than the random baseline.” This confusion may have been caused by the fact that higher AUC indicates a poorer attribution method for ROAR while it indicates a better attribution method for KAR.
> >
> > We address the Reviewer’s comments on Sensitivity and ROAR in the answer to the next comment.

---

> > > ### Author Response · Authors · 2018-11-16
> > > **Reply to AnonReviewer3 Part 3/3**
> > >
> > > C6: "The authors mentioned that this (worse performance of RectGrad on ROAR and Sensitivity) may be because of the sparsity of RectGrad. However, if the sparsity is the harm, the underlying observations of RectGrad may have some errors. I think the current manuscript has an inconsistency between the fundamental idea (based on empirical observations) and the performance of RectGrad."
> > >
> > > A6: The result on Sensitivity does not contradict but corroborates our claims on RectGrad. As we now explain in Section 4.1 (Rationale Behind the Propagation Rule for RectGrad), RectGrad theoretically (1) removes noise and (2) thresholds out features by order of importance (note that (1) is a consequence of (2)). The Sensitivity result shows that the logit when the features are occluded according to RectGrad for the first few patches drops faster than or as fast as those of other methods. This indicates that RectGrad has successfully captured the most important features, as we have claimed.
> > >
> > > We believe the drop in performance after the first few patches is an inevitable consequence of sparsity. To see this, we have performed an additional experiment where we apply final threshold to baseline attribution methods so that RectGrad attribution maps and baseline attribution maps have similar levels of sparsity. We observed that at similar levels of sparsity, there is no significant difference between the performance of RectGrad and baseline methods.
> > >
> > > Given that attribution maps are ultimately used by humans to “visually” interpret DNN decisions, we believe sparsity / visual quality is also an important factor in DNN interpretability. Noisy attribution maps such as “lighter” example in Figure 6 can hinder the user’s attempt to interpret DNN decisions. As our experiments show, baseline attribution maps are still noisy after final threshold. On the same level of sparsity, RectGrad attribution maps are significantly less noisy but still better than or as good as baseline attribution maps in highlighting important features (results described in reply to AN1, A5). Hence we believe RectGrad has a notable advantage over baseline attribution methods.
> > >
> > > As for the explanation for ROAR in Appendix B.2, our intention was to show that this metric may not be suitable for objectively evaluating RectGrad. To explain why, we cite a sentence from Hooker et al. (2018, https://arxiv.org/abs/1806.10758) which proposed ROAR: “training the model from random initialization is crucial in order for the constant value for which we replaced the input to be considered “uninformative””. The assumption behind ROAR is that the occluded features do not influence the classification task. However, as we have shown in Appendix B.2, this does not seem to be the case for RectGrad.
> > >
> > > [Minor Concern]
> > >
> > > C7: “In Sec.5, the authors frequently refer to the figures in appendix. I think the main body of the paper should be self-contained. I therefore think that some of the figures related to main results should appear in the main part.”
> > >
> > > A7: We were notified that ICLR had set a strict limit on paper length after drafting. We tried our best to insert as much figures related to the main results as possible into the main part. Moving more figures will cause the paper to violate the 10-page limit. We also decided to insert qualitative experiment figures in the main part since it is relatively easier to accurately describe quantitative results with words.

---

> > > > ### Comment · AnonReviewer3 · 2018-11-29
> > > > **Sec4.1 is still not convincing.**
> > > >
> > > > I think the discussion in Sec4.1 is still not appropriate for justifying RectGrad.
> > > > Suppose that we have f(x, y, z) = -100x + 2y + z, and suppose that we feed x=3, y=2, z=1. Then, it is apparent that the value of f is dominated by -100x. Thus, we expect x to be selected as the most influential variable. However, in RectGrad, because -100x < 0, the gradient is filtered out, and thus the contribution of x is evaluated as zero. On the other hand, because 2y > z > 0, the variable y is found to be most influential.
> > > > In this case, x can be found as the most influential variable if we adopt I(a>t) * R.

---

> > > > > ### Author Response · Authors · 2018-11-30
> > > > > **Re: Sec4.1 is still not convincing.**
> > > > >
> > > > > In the reply, the Reviewer defines the most influential variable as one which dominates the function output. We claim that RectGrad propagation rule I(a * R > t) * R indeed correctly identifies the most influential, or according to the Reviewer, the dominating variable. To show this, we start by pointing out a flaw in the Reviewer’s argument: I(a > t) * R does not always select the dominating variable.
> > > > >
> > > > > Suppose we have f(x, y, z) = -100x + 1000y + z and feed (x, y, z) = (3, 2, 1). Here the function output is dominated by 1000y. Under the Reviewer’s logic, y is the most influential variable since it dominates. However, because y < x, x is selected to be the most influential variable under I(a > t) * R. Therefore, instead of I(a > t) * R, we should use I(|a * R| > t) * R if we are to naively choose the dominating variable. This propagation rule selects the dominating variable in both examples. In fact, this propagation rule selects the dominating variable in any linear model since |a * R| is the absolute value of the input multiplied by its weight.
> > > > >
> > > > > Reviewer’s Example: f(x , y, z) = -100x + 2y + z with (x, y, z) = (3, 2, 1).
> > > > >
> > > > > Our Counterexample: f(x, y, z) = -100x + 1000y + z with (x, y, z) = (3, 2, 1).
> > > > >
> > > > > However, there is a problem with this propagation rule. Recall that in a typical multi-class classification setting, the class with the *largest* logit is selected as the decision of the network. Hence it is logical to define important units as those with the largest contribution (a * R), not the largest absolute contribution (|a * R|). For instance, in the Reviewer’s example, even though -100 x dominates with the largest absolute contribution, it contributes least to the output due to its negative sign.
> > > > >
> > > > > Hence a reasonable propagation rule should first identify the units which have the largest contribution (a * R) to the output and then select the dominating unit(s) among them. At this point, it is evident that the rule which satisfies this condition is I(|a * R| > t) * R without the absolute value: I(a * R > t) * R. This is the RectGrad propagation rule.
> > > > >
> > > > > In similar but different contexts, Ancona et al. (2018, https://arxiv.org/abs/1711.06104), Smilkov et al. (2017, https://arxiv.org/abs/1706.03825), Sundararajan et al. (2017, https://arxiv.org/abs/1703.01365), and Shrikumar et al. (2017, https://arxiv.org/abs/1704.02685) have discussed the implications and advantages of multiplying activation with gradient. Especially, Ancona et al. shows in Section 3.2 that activation * gradient “should be used to identify the marginal effect that the presence of a feature has on the output, which is usually desirable from an explanation method.”

---

> > > > > > ### Comment · AnonReviewer3 · 2018-11-30
> > > > > > **The claim seems to be correct only for the last layer.**
> > > > > >
> > > > > > I agree that I(|a * R| > t) * R is a reasonable choice for linear models.
> > > > > > However, relaxing I(|a * R| > t) * R to I(a * R > t) * R is still questionable for general deep models. If we focus on the last logit layer, the relaxation seems to be reasonable because we are interested in argmax. However, the proposed RectGrad is applied also to middle layers where there is no reason to prefer a*R instead of |a*R|.

---

> > > > > > > ### Author Response · Authors · 2018-12-01
> > > > > > > **Re: The claim seems to be correct only for the last layer.**
> > > > > > >
> > > > > > > We give both intuitive and rigorous explanations as to why we should use a * R instead of |a * R|.
> > > > > > >
> > > > > > > Intuitive explanation: Since |a * R| does not work for even the simplest examples, it is highly likely that this will not work DNNs which are constructed by composing multiple affine layers. On the other hand, for a * R, we can iteratively apply our reasoning for the linear model case at each layer. Then a * R selects units with the largest marginal effect in a layer-wise manner.
> > > > > > >
> > > > > > > Rigorous explanation: Suppose we have an L-layer DNN. We show by induction that RectGrad identifies units having the largest contribution to the output at all layers.
> > > > > > >
> > > > > > > Base case: We have already shown that the claim is true for the base case in the previous comment. That is, given the output at layer L, RectGrad correctly identifies important units at layer L – 1.
> > > > > > >
> > > > > > > Inductive step: In this step, we show that if RectGrad correctly identifies influential units at layer k, then we can again apply RectGrad to identify influential units at layer k – 1. Suppose we are at an arbitrary hidden layer. This layer can be modeled as a vector-valued affine function f mapping R^n to R^m. Then f(v) = ReLU(f_1(v), ..., f_m(v)) where v is a vector in R^n and each f_i are scalar-valued affine functions. By induction hypothesis, RectGrad identifies the functions f_i which have the largest contribution to the output. Then RectGrad assigns gradient R_i to functions with the largest contribution and 0 to others. Note that R_i is an approximate measure of how sensitive the output is to changes in f_i. Specifically, the output is roughly c + R_i * f_i(v) for an appropriate constant c.
> > > > > > >
> > > > > > > Now we show RectGrad correctly identifies the influential units in v. Without loss of generality, suppose f_1 is identified as the unit with the largest contribution to the output. Then (output) ~ c + R_1 * f_1(v). To demonstrate that |a * R| does not work, we reuse the Reviewer’s toy example f_1(v) = f(x, y, z) = -100x + 2y + v:
> > > > > > >
> > > > > > > (output) ~ c + R_1 * (-100x + 2y + z) with (x, y, z) = (3, 2, 1).
> > > > > > >
> > > > > > > Now let us compare  |a * R| and a * R:
> > > > > > >
> > > > > > > x / a = x / R = -100 R_1 / a * R = -100 R_1 x = -300 R_1 / |a * R| = |-300 R_1|
> > > > > > > y / a = y / R = 2 R_1       / a * R = 2 R_1 y      = 4 R_1       / |a * R| = |4 R_1|
> > > > > > > z / a = z / R = R_1          / a * R = R_1 z          = R_1          / |a * R| = |R_1|
> > > > > > >
> > > > > > > If R_1 > 0, x still contributes least to the output due to its negative sign. However, it has the largest |a * R| value. Therefore using I(|a * R| > t) * R may cause the gradient to be propagated through units with the least contribution to the output. Now, observe that
> > > > > > >
> > > > > > > (output) ~ c + R_1 * (-100x + 2y + z) = c + (a * R for x) + (a * R for y) + (a * R for z).
> > > > > > >
> > > > > > > Hence I(a * R > t) * R correctly identifies influential units since a * R is approximately the amount of the unit’s contribution to the output. Clearly this reasoning applies even when multiple units f_i are identified as having the largest contribution to the output since a linear combination of affine functions is still affine:
> > > > > > >
> > > > > > > (output) ~ c + R_1 * f_1(v) + ... + R_k * f_k(v) = c + g(v) = c’ + (a * R for a_1) + ... + (a * R for a_k)
> > > > > > >
> > > > > > > where g = R_1 * f_1(v) + ... + R_k * f_k(v).
> > > > > > >
> > > > > > > Since we have shown the base case and the inductive step, our claim holds for all layers.
> > > > > > >
> > > > > > > In addition, we have generated a few samples comparing RectGrad with the modified propagation rule which uses I(a * R) * R for the final layer and I(|a * R|) * R for hidden layers. We uploaded the samples in an anonymous Google drive:
> > > > > > >
> > > > > > > https://drive.google.com/drive/folders/1jXjSrrgFH60uyNORDO2pyncYeWUhLmGE?usp=sharing
> > > > > > >
> > > > > > > We have observed that the modified propagate rule often fails to highlight discriminating features of the object of interest or highlights the background (e.g. “Carton” or “Soup bowl” example). This corroborates our claim that since |a * R| does not work for even the simplest examples, it is highly likely that this will not work DNNs which are constructed by composing multiple affine layers.

---

### Official Review · AnonReviewer1 · 2018-11-02
**This paper proposes a new method for producing saliency maps and my main concerned is the objective evaluation.**

**Rating:** 5
**Confidence:** 4

**Review:**

This paper studies how to better visually interpret a deep neural network. It proposes a new method to produce less noisy saliency maps, named RectGrad. RectGrad thresholds gradient during backprop in a layer-wise fashion in a similar manner to a previous work called Guided Backprop. The difference is that Guided Backprop employs a constant threshold, i.e. 0, while RectGrad uses an adaptive threshold based on a percentile hyper-parameter. The paper is well-written, including a comprehensive review of previous related works, an meaningful meta-level discussion for motivation, and a clear explanation of the proposed method.

One of my biggest concern is regarding the experiment and evaluation section. Conclusions are drawn based on the visualization of a few saliency maps. I am not sure how much I can trust these conclusions as the conclusions are drawn in a handy-wavy manner the examples are prone to cherry-picking and . For example, this is the conclusion in the Adversarial Attack paragraph: “we can conclude that Rectified Gradient is  equally or more class sensitive than baseline attribution methods”. As pointed out by the paper, the conclusion can be drawn from Figure 8 in the main paper and Figure 10 in Appendix A.1. However, the proposed method tends to produce a saliency map with higher sparsity, therefore the difference may appear more apparent. It is stretching to conclude that it is more class sensitive without further quantitative validation.

Evaluation appears to be a common concern to the work on saliency maps.  The existing quantitative evaluation in the paper seems disconnected to the visual nature of saliency maps. Concretely, when can we say one saliency map looks better than another? Since this paper claims to produce less noisy saliency maps, what does it mean quantitatively? Is it true that it produces less pixels on the background? If so, can we evaluate it with foreground-background segmentation annotation to prove that point? Though how to evaluate saliency maps remains an open question, I feel some discussion on this paper would make the paper more insightful.

---

> ### Author Response · Authors · 2018-11-16
> **Reply to AnonReviewer1 Part 1/2**
>
> Thank you for the friendly and detailed review. Before reading our reply for your review, we politely ask you to read “Our Common Reply to All Reviewers” first.
>
> C1: "One of my biggest concern is regarding the experiment and evaluation section. Conclusions are drawn based on the visualization of a few saliency maps. I am not sure how much I can trust these conclusions as the conclusions are drawn in a handy-wavy manner the examples are prone to cherry-picking. For example, this is the conclusion in the Adversarial Attack paragraph: “we can conclude that Rectified Gradient is equally or more class sensitive than baseline attribution methods”. As pointed out by the paper, the conclusion can be drawn from Figure 8 in the main paper and Figure 10 in Appendix A.1."
>
> A1: We have surveyed samples for 1.5k randomly chosen ImageNet images, and we found them to be generally consistent with our claims in the paper. As we mentioned in “Our Common Reply to All Reviewers”,  we have uploaded the 1.5k samples in anonymous Google drives so that the Reviewers can inspect them if he/she is not convinced by the results in the paper. However, as it always is with DNN interpretability research, it is difficult to demonstrate the efficacy of the proposed method or erase the cherry-picking concern just with large-scale samples. We are aware of this fact and that is why we have conducted numerous additional quantitative experiments (e.g. AN2 A3). It would be very much appreciated if the Reviewer understands and takes this situation into account when reviewing the revised version of this work.
>
> C2: "However, the proposed method tends to produce a saliency map with higher sparsity, therefore the difference may appear more apparent."
>
> A2: We applied final threshold to baseline attribution methods such that RectGrad and baseline attribution maps have similar levels of sparsity. The final threshold details are described in Our Common Reply to All Reviewers. We then repeated all qualitative experiments on 1.5k randomly chosen ImageNet images (image links listed in comment above). We have observed that the conclusions still generally hold even after final threshold.
>
> We also believe that the ability of our method to control the threshold hyper parameter to make the difference more/less apparent is an advantage, rather than a disadvantage. If the image was near the decision boundary in the first place, then the adversarial attack would change only a small portion of internal activation patterns. This may lead to attribution maps which are indistinguishable from attribution maps for original images. Baseline methods have no effective way of dealing with this except final threshold. However, as we have observed, final threshold still results in noisy attribution maps for baseline methods. For RectGrad, the user can control the threshold percentile q to visually understand which features really account for the change in DNN decision.
>
> C3: "It is stretching to conclude that it is more class sensitive without further quantitative validation."
>
> A3: To our knowledge, there is no previous work in quantitatively validating how class sensitive an attribution method is. There only exist qualitative methods: (1) comparing original attribution map with those produced with respect to another class (Smilkov et al. 2017, https://arxiv.org/abs/1706.03825) and (2) adversarial attack (Nie et al. 2018, https://arxiv.org/abs/1805.07039). We have chosen the latter method since it is deterministic; for the former method, there are 1000 other classes to choose from, and we thought this could lead to space trouble or cherry-picking concerns. Hence the large-scale adversarial attack samples are the best we can do to convince the Reviewer in the current situation.

---

> > ### Author Response · Authors · 2018-11-16
> > **Reply to AnonReviewer1 Part 2/2**
> >
> > C4: "The existing quantitative evaluation in the paper seems disconnected to the visual nature of saliency maps."
> >
> > A4: We assumed that the samples inserted in the paper would be enough to convince the Reviewers that RectGrad produces attribution maps with significantly less noise than baseline methods. That is why the quantitative experiments are focused on evaluating whether RectGrad attribution maps are truly informative, i.e., highlights features relevant to DNN decision. However the Reviewer does not seem to be convinced due to the cherry-picking concern. We address this concern in “Our Common Reply to All Reviewers.” If the Reviewer is still not convinced, we conducted additional quantitative experiments to prove that RectGrad significantly reduces noise in saliency maps. We summarize the results in the reply below.
> >
> > C5: "Since this paper claims to produce less noisy saliency maps, what does it mean quantitatively? Is it true that it produces less pixels on the background? If so, can we evaluate it with foreground-background segmentation annotation to prove that point?"
> >
> > A5: If the reviewer is not convinced by “Our Common Reply to All Reviewers”, we performed two additional quantitative experiments to prove that RectGrad reduces noise.
> >
> > First, as the Reviewer suggested, we created segmentation masks for 10 correctly classified CIFAR10 images of each class (total 100 images) and measured how much attribution falls on the background. Specifically, we compared the sum of absolute value of attribution on the background. Figure 13 shows that the RectGrad assigns significantly less attribution to the background than baseline methods. Moreover, even with final threshold (so that RectGrad and baseline attribution maps have similar sparsity), RectGrad outperformed baseline methods.
> >
> > We also measured how noisy each attribution maps are using the total variation metric, and we believe the results can further support our reply to C5. To measure how noisy attribution maps were before and after the final threshold, we measured the total variation of each attribution map and took the average across the test dataset. Figure 14 shows that even though the total variation reduces for baseline methods after final threshold, RectGrad outperforms baseline methods in both cases.
> >
> > To summarize, we have shown through (1) foreground-background segmentation annotation and (2) total variation metric that RectGrad attribution maps are significantly less noisy than baseline attribution maps both with and without final threshold.
> >
> > C6: "Evaluation appears to be a common concern to the work on saliency maps." + "Though how to evaluate saliency maps remains an open question, I feel some discussion on this paper would make the paper more insightful." + "Concretely, when can we say one saliency map looks better than another?"
> >
> > A6: We agree that attribution map evaluation is an important line of DNN interpretability research. However, we disagree with the Reviewer’s comment that “some discussion on this paper would make the paper more insightful.” The main contributions of this paper are (1) identifying why saliency maps are noisy and (2) proposing a solution. Proposal or discussion of evaluation metric for attribution methods are out of scope of this paper. Hence we believe that adding such content will cloud the focus of our paper.
> >
> > Lastly, we are worried about the fact that the Reviewer did not mention Section 3 (Our Explanation for Saliency Maps) which we included in the list of our contributions. We believe this contribution is significant in our work for the following reason: this paper thoroughly investigates and proposes an answer to the question of why saliency maps are noisy. To the best of our knowledge, this question has never been answered previously. There are several studies which propose hypotheses (e.g. Smilkov et al. 2017, https://arxiv.org/abs/1706.03825; Shrikumar et al. 2017, https://arxiv.org/abs/1704.02685), but they do not conduct extensive experiments to corroborate their claims. Thus, we hope the Reviewer takes this contribution into account when reviewing the revised version of this paper.

---

### Official Review · AnonReviewer2 · 2018-11-03
**Interesting work, but I believe a few questions need to be answered to make the paper strong enough for acceptance.**

**Rating:** 5
**Confidence:** 5

**Review:**

Summary of the paper:
This paper proposed RectGrad, a gradient-based attribution method that tries to avoid the problem of noise in the attribution map. Further, authors hypothesize that noise is caused by the network carrying irrelevant features, as opposed to saturation, discontinuities, etc as hypothesized by related papers.

The paper is well written and easy to read through.

Strengths:
- Formally addresses a hitherto unanswered question of why saliency maps are noisy. This is an important contribution.
- RectGrad is easy to implement.

Questions for authors:
- Since the authors are saying that the validity of their hypothesis is “trivial”, it would be nice to have this statement supported by more quantitative, dataset-wide analyses on the feature map and training dataset occlusion tests. For e.g., what percentage of the test dataset shows attributions on the 10x10 occluded patch?
- How does RectGrad compare with simply applying a final threshold on other attribution maps? How do the results on training data and feature occlusion change after such a threshold is applied? How do results on adversarial attacks change?
- Could this method generalize to non-ReLU networks?
- Premise that auxiliary objects in the image are part of the background is not necessarily true. For instance, the hand in “lighter” is clearly important to know that the flame is from a lighter and not from a candle or some other form of fire. Similarly, the leaves in the “frog” example.
- (Optional) As shown in (https://openreview.net/forum?id=B1xeyhCctQ) gradients on ReLU networks overlook the bias term. In the light of this, what is the authors’ take on whether a high bias-attribution is the cause for the noisy gradient-attribution?
- (Optional) In some sense, RectGrad works because layers closer to the input may capture more focussed features than layers close to input which may activate features spread out all over the image. It would be interesting to see if RectGrad works for really small networks such as MobileNet (https://arxiv.org/abs/1801.04381) where such an explicit hierarchy of features may not be there.

---

> ### Author Response · Authors · 2018-11-16
> **Reply to AnonReviewer2 Part 1/3**
>
> Thank you for the friendly and detailed review. Before reading our reply for your review, we politely ask you to read “Our Common Reply to All Reviewers” first.
>
> C1: "Since the authors are saying that the validity of their hypothesis is “trivial”, it would be nice to have this statement supported by more quantitative, dataset-wide analyses on the feature map and training dataset occlusion tests. For e.g., what percentage of the test dataset shows attributions on the 10x10 occluded patch?"
>
> A1: Before summarizing additional experiment results, we would like to make a clarification. Our hypothesis is comprised of two parts: (1) background features cause noise in saliency maps and (2) background features are trivial or irrelevant to the classification task. The Reviewer seems to be implying that we have claimed both parts (1) and (2) to be trivial. However, it is only part (1) that we have claimed to be trivial by the definition of gradient. We have never claimed part (2) is trivial.
>
> We performed two additional experiments to demonstrate that DNNs do not filter out irrelevant features during forward propagation / that background feature activations are irrelevant to the classification task.
>
> First, as the Reviewer suggested, we measured how much attribution is on the 10x10 occluded patch. Since we don’t have a criterion of how much attribution is trivial enough to be seen as “no attribution”, we instead summed all absolute attribution within the patch and took the average across the test dataset. We repeated this with other attribution methods and created a bar chart comparing the average. Results are shown in Figure 8. Ideally, there should be nearly zero attribution, but we observed that the saliency map assigned the most attribution to the patch among all attribution methods. This shows that DNNs do not filter our irrelevant features during forward propagation (by definition of gradient) and that other attribution methods alleviate this problem.
>
> Second, we created segmentation masks for 10 correctly classified CIFAR10 images of each class (total 100 images) and repeated the feature map occlusion test. We recorded (class logit) – (largest logit among the other 9 classes) and took the average over all the images. Figure 9 in Appendix A.1 shows that the difference is generally positive throughout the occlusion process (i.e., the class does not change throughout the occlusion process), and this implies the irrelevance of background features to the classification task.
>
> C2: "How does RectGrad compare with simply applying a final threshold on other attribution maps?"
>
> A2: We found that noise can accumulate during backpropagation. Specifically, irrelevant features may have trivial gradient near the output layer; however, since gradient is calculated by successive multiplication, the noise can grow exponentially as gradient is propagated towards the input layer. This often results in confusing attribution maps which assign high attribution to entirely irrelevant regions (e.g. baseline methods assign high attribution to uniform background in "lighter" example in Figure 6), especially for deep networks such as Inception. In such situation, simply applying a final threshold does not work. RectGrad does not suffer from this problem since it thresholds irrelevant features at every layer and hence stops noise accumulation in the first place (Section 4.1 explains how RectGrad thresholds irrelevant features).
>
> (Updated 11/23) In Appendix A.2, we corroborate this claim by comparing Saliency map and RectGrad attributions as they are propagated towards the input layer.
>
> We have also surveyed samples for 1.5k randomly chosen ImageNet images, and we found them to be generally consistent with our claims in the paper. As we mentioned in “Our Common Reply to All Reviewers”,  we have uploaded the 1.5k samples in anonymous Google drives so that the Reviewers can inspect them if he/she is not convinced by the results in the paper. However, as it always is with DNN interpretability research, it is difficult to demonstrate the efficacy of the proposed method or erase the cherry-picking concern just with large-scale samples. We are aware of this fact and that is why we have conducted numerous additional quantitative experiments (e.g. A3). It would be very much appreciated if the Reviewer understands and takes this situation into account when reviewing the revised version of this work.

---

> > ### Author Response · Authors · 2018-11-16
> > **Reply to AnonReviewer2 Part 2/3**
> >
> > C3: "How do the results on training data and feature occlusion change after such a threshold is applied? How do results on adversarial attacks change?"
> >
> > A3: The final threshold setting is described in “Our Common Reply to All Reviewers”.
> >
> > For the training data occlusion test, we created another bar chart summarizing the results after applying a final threshold (test and bar chart details are described in A1). For this test, we found using q = 95 final threshold led to trivially different averages. Hence we used a custom threshold for each baseline method such that they had similar average attribution in the patch as RectGrad. Figure 8 shows that RectGrad had smaller standard deviation than baseline methods. This indicates that RectGrad more consistently assigns near-zero attribution to the patch. Therefore, RectGrad has advantages over baseline methods regardless of whether final threshold is used or not.
> >
> > For “feature occlusion”, we assume it is the Sensitivity experiment in Section 5, not feature map occlusion experiment in Section 3. We observed that after applying the final threshold, RectGrad still outperforms local attribution methods. RectGrad initially performed worse than global attribution methods after occluding approx. 10 patches. However, after final threshold, RectGrad now shows similar performance.
> >
> > This was not requested by the Reviewer, but we measured how noisy each attribution maps are using the total variation metric, and we believe the results can further support our reply to C3. To measure how noisy attribution maps were before and after the final threshold, we measured the total variation of each attribution maps and took the average across the test dataset. Figure 14 shows that even though the total variation reduces for baseline methods after final threshold, RectGrad outperforms baseline methods in both cases.
> >
> > Finally, we applied final threshold to baseline attribution maps for adversarial attack images. Figures 7 and 12 show the results. We found that our observations still held (that is, RectGrad is more or as class-sensitive as baseline methods). Also, the baseline attribution maps with final threshold were still noisy.
> >
> > With the above four additional experiments, we can draw the following conclusions:
> >
> > (1) RectGrad more consistently assigns zero attribution to irrelevant features.
> > (2) At the same level of sparsity, RectGrad performs better than or similar to baseline methods in Sensitivity. This shows that RectGrad is better than or as good as other methods in selecting important features.
> > (3) At the same level of sparsity, RectGrad is more or as class-sensitive as baseline methods.
> > (4) At the same level of sparsity, RectGrad attribution maps are significantly less noisy than baseline attribution maps.
> >
> > (Conclusion 4 is quantitatively supported by A5 of our reply to AN1 where we answer "Since this paper claims to produce less noisy saliency maps, what does it mean quantitatively? Is it true that it produces less pixels on the background? If so, can we evaluate it with foreground-background segmentation annotation to prove that point?")
> >
> > Given that attribution maps are ultimately used by humans to “visually” interpret DNN decisions, we believe sparsity / clarity is also an important factor in DNN interpretability. Noisy attribution maps such as “lighter” example in Figure 6 can hinder the user’s attempt to interpret DNN decisions. As our experiments show, baseline attribution maps are still noisy after final thresholding. On the same level of sparsity, RectGrad attribution maps are significantly less noisy but still better than or as good as baseline attribution maps in highlighting important features. Hence we believe RectGrad has a notable advantage over baseline attribution methods.
> >
> > C4: "Could this method generalize to non-ReLU networks?"
> >
> > A4: We direct the Reviewer to Section 4.1, where we explain the rationale behind the definition of RectGrad. The reasoning that RectGrad propagates gradient through units whose marginal effect on the output exceeds some thresholds applies to any DNN network structure, regardless of the type of activation function used. We corroborate this claim with experiment results on CIFAR10 with Sigmoid and TanH  DNNs. We used the same architecture as ReLU DNN and trained each for 40 epochs to achieve 70.5% and 75% test accuracy respectively. We observed that RectGrad PRR does not work as well as vanilla RectGrad for Sigmoid and TanH.
> >
> > Please note that these results and discussions are not included in the current version of revised paper due to the 10-page limit. However, if the Reviewer feels this result is important enough, we will insert them in the Appendix.
> >
> > Sigmoid
> > https://drive.google.com/drive/folders/1c_qLKm-uOB-Dcz5KVpvAnlFPHZL9I6FH?usp=sharing
> >
> > TanH
> > https://drive.google.com/drive/folders/18bMsizZ-geHqMQp8pGSHbXGjJk8f1Pms?usp=sharing

---

> > > ### Author Response · Authors · 2018-11-16
> > > **Reply to AnonReviewer2 Part 3/3**
> > >
> > > C5: "Premise that auxiliary objects in the image are part of the background is not necessarily true."
> > >
> > > A5: We agree with the reviewer that auxiliary objects may influence the DNN decision process. That is why we have the hyper parameter threshold percentile q so that the user can control the degree to which RectGrad emphasizes important features. We direct the reviewer to Figure 5 where we explore the effect of varying q. Attribution maps with q = 80 ~ 90 highlights the object of interest (the cabbage butterfly) along with auxiliary objects such as flowers or grass that may be helpful to the DNN in identifying the object. On the other hand, attribution maps with q > 95 highlight features that may have been most influential to the final decision, namely the spots on the butterfly's wing. Auxiliary objects may not have been highlighted in the RectGrad attribution maps in the paper since we used a high threshold q = 98.
> > >
> > > C6: "For instance, the hand in “lighter” is clearly important to know that the flame is from a lighter and not from a candle or some other form of fire. Similarly, the leaves in the “frog” example."
> > >
> > > A6: For the “lighter” example, the attribution also highlights the cap/hood of the lighter which indicates that RectGrad correctly explains how the DNN distinguished the lighter flame from other forms of fire. Also lowering the threshold parameter q to 90 also highlights the hand. For the “frog” example, we found that RectGrad attribution highlighted the leaves only slightly for q = 80, which implies leaves may be irrelevant or trivial to the classification task. This explanation is plausible since other methods also assign relatively small attribution to the leaves.
> > >
> > > Lighter image: https://drive.google.com/open?id=1mj0H5hdXUQRrcT0J-AJximd2eYfJhEIh
> > >
> > > Frog image: https://drive.google.com/open?id=1VBxiO8iO-cKxwFvc5Fkw9r8sXTLSbRqK

---

### Author Response · Authors · 2018-11-16
**Changes to the Paper**

(1) Added Section 4.1 which explains the rationale behind the propagation rule for RectGrad, moved Section 4.1 (Relation to Deconvolution and Guided Backprop.) to Section 4.2, and moved Section 4.2 (Useful techniques) to Appendix.

(2) Added Appendix A.1 which describes the procedure and results of a larger-scale version of the feature map occlusion experiment, moved Appendix A.1 (Qualitative Experiments) to A.2, and moved Appendix A.2 (Quantitative Experiments) to A.3.

(3) Moved “Training Dataset Occlusion” test from Section 5.2 (Qualitative Comparison with Baseline Methods) to Section 5.3 (Quantitative Comparison with Baseline Methods).

(4) Added "Noise Level" test in Section 5.3 (Quantitative Comparison with Baseline Methods).

(5) Added experimental results comparing RectGrad with baseline methods with final threshold.

(6) Revised main text to reflect Reviewer comments.

(7) Resized Figure 3 and removed redundant subfigures from Figure 4.

(8) Changed in-text reference to the proposed method from “Rectified Gradient” to “RectGrad”.

(9) Removed results on MNIST as we felt they added little to the paper.

(10) Moved attribution map visualization method to Appendix F.1.

(Updated 11/23) (11) Added Appendix A.2, which experimentally corroborates the claim that RectGrad prevents noise accumulation through importance score based thresholding.

---

### Author Response · Authors · 2018-11-16
**Our Common Reply to All Reviewers**

We thank the three reviewers for providing detailed and constructive feedback that will no doubt help us improve the quality of this work.

We addressed the reviewers’ comments in detail individually. We number the comments by ”C{number}” and answers by “A{number}” for ease of reference. We also refer to AnonReviewer{number} by “AN{number}”. We also uploaded our revised manuscript where we have reflected the reviewers’ comments and feedback.

The figures that we refer to in the answers are those in the revised version of the paper, not the original version. Figures have been added, removed, or modified during the revision process. Hence using figures in the original paper may lead to confusion.

To address the general concern about cherry-picking, we have repeated the qualitative experiments for 1.5k randomly chosen ImageNet images (both before final thresholding and after final thresholding). We believe that these results alleviate the cherry-picking concern.

There also has been repeated (explicit and implicit) questions on whether applying simple final threshold to baseline attribution maps is enough to replicate the benefits of RectGrad. To refute this concern, we applied 95 percentile final threshold to baseline attribution methods such that RectGrad and baseline attribution maps have similar levels of sparsity. Note that we did not apply the threshold q = 98, which was used in our RectGrad results. In the setting of q = 98 on baseline methods, RectGrad attribution maps are slightly less sparse than baseline attribution maps. This is because threshold is applied up to the first hidden layer, not the input layer in the RectGrad procedure. With this final threshold on baseline methods, we repeated the qualitative and quantitative experiments . We also repeated the qualitative experiments for the same 1.5k randomly chosen ImageNet images.

The links to anonymized Google drives containing the 1.5k random samples are listed in the comment below.

Please let us know if the Reviewers have any further questions or comments.

---

> ### Author Response · Authors · 2018-11-16
> **Links to Samples on 1.5k Randomly Chosen ImageNet Images**
>
> Note1: Samples for the same image have the same file name “{image number}.png” for ease of comparison between attribution maps with and without threshold / attribution maps for original image and adversarial image.
>
> Note2: We did not include samples for adversarial attacks which failed to change the final decision.
>
> Random Samples W/O Baseline Final Threshold
> https://drive.google.com/drive/folders/1F9k-Jvxe1OppDoIDHV1SWLzugkpPQMkY?usp=sharing
>
> Random Samples with Baseline Final Threshold
> https://drive.google.com/drive/folders/1LQOWtvJPV9nnUCjB2VRNDgFGixmpAnJB?usp=sharing
>
> Adversarial Attack Samples W/O Baseline Final Threshold
> https://drive.google.com/drive/folders/1kkgVQs2jBWWqLW5CTrKhpBnpwAUhZDJs?usp=sharing
>
> Adversarial Attack Samples with Baseline Final Threshold
> https://drive.google.com/drive/folders/1MXrmQCgWgpf84Mj1f9Q8QrOfOncSbvlf?usp=sharing

---

### Meta-Review · Area_Chair1 · 2018-12-15

**Confidence:** 4
**Recommendation:** Reject

**Metareview:**

The main goal of the submission is to figure out a way to produce less "noisy" saliency maps. The RectGrad method uses some thresholding during backprop, like Guided Backprop. The visuals of the proposed method are good, but the reviewers rightfully point out that evaluating whether the proposed method is any good is not obvious. The ROAR/KAR results are perhaps not telling the whole story (and the authors claim that RectGrad is not expected to get a high ROAR score, but I would like to see this developed more in a further version of this work).

Generally, I feel like there was a healthy back and forth between authors and R3 on the main concerns of this work. I agree that the mathematical justification for RectGrad seems not fully developed. Given all of these concerns, at this point I cannot support acceptance of this work at ICLR.